# The new normal: Covid-19 risk perceptions and support for continuing restrictions past vaccinations

**Maja Graso** [ID] *

Department of Management, University of Otago, Dunedin, New Zealand

* majagraso@gmail.com

**Data Availability Statement:** All anonymized data and syntax with basic instructions are available here: https://osf.io/87dzs/?view_only=ef9c8daf19c74e078c1c2d2abd3a06e0.

## Abstract

I test the possibility that over-estimating negative consequences of COVID-19 (e.g., hospitalizations, deaths, and threats to children) will be associated with stronger support the '*new normal*' (i.e., continuation of restrictions for an undefined period starting with wide-spread access to vaccines and completed vaccinations of vulnerable people). The *new normal* was assessed by endorsing practices such as vaccine passports, travel restrictions, mandatory masking, continuing contact tracing, and pursuing elimination. Results are based on five samples (*N* = 1,233 from April 2021 and *N* = 264 from January 2022) and suggest that people *over*-estimate COVID-19 risks to children and healthy people, as evidenced by median estimates that 5% of all global deaths were children, 29% were generally healthy people under 65, and that a healthy person under the age of 65 has 5% chance of dying from COVID-19. Over-estimates observed in this study align with those based on representative samples, and they were consistently related to stronger support for the new normal. This relationship emerged when participants estimated risks with percentages (core indicators) and indicated the extent to which risk-based statements are true/supported with evidence or false/unsupported (alternative indicators). People were notably more likely to support continuing restrictions if they believed that COVID-19 risk and risk mitigation tactics are true, even when they are not (e.g., children need to be prioritized for boosters). These relationships persisted when considering competing explanations (political ideology, statistics literacy, belief in conspiracy theories). I trace these effects to well-meaning efforts to prevent under-estimation. Public policy and people's perceptions of risks are intertwined, where even inaccurate judgments may influence decisions. Failure to combat *all* misinformation with equal rigor may jeopardize the restoration of the social and economic life essential for building adaptive post-pandemic societies.

## Introduction

The nearly universal desire to 'flatten the curve', prevent hospitals from becoming overwhelmed, and save lives mobilized millions to embrace numerous health-minded practices

**Funding:** Yes. This study is funded by the University of Otago's internal grant system (University of Otago Research Grant). The funders had no role in study design, data collection and analysis, decision to publish, or preparation of the manuscript.

**Competing interests:** The author has declared that no competing interests exist.

and protect themselves and their fellow citizens from COVID-19. The end of the pandemic now looks closer than ever, thanks to vaccines that are continuing to be effective at preventing severe illnesses and deaths from COVID-19 [1, 2]. With the worst of the risks waning, signs of a return to pre-pandemic times are rising. Some places are dropping most of their COVID-19 restrictions entirely (e.g., Florida, Texas, and the UK), and others are adjusting in tandem with local conditions.

Risk, however, is not constant, nor can it be reduced to a binary variable where the outcome is either certain death or insulation from all harm [3–5]. COVID-19 comes with gradients of risk due to new variants, continuously rising cases across the globe, varying levels of vaccinations across states and countries, and the uncertainty of long-term effects of infections [6]. Despite their high effectiveness, COVID-19 vaccines–like any preventative measure–do not eliminate the risk [1]. Some COVID-19 risks will likely remain until the virus is no longer seen as a threat because people are protected from severe illnesses [7] or is eliminated through other strict responses [8].

Despite vaccines not being a 'magic bullet' to end the pandemic [8], wide-spread vaccination uptakes, particularly of all vulnerable people who are more susceptible to needing hospital care, offer a timely opportunity to re-evaluate the cost/benefit analysis of the severe restrictions, and to consider the public's role in this process. Continuing or re-implementing restrictions despite high vaccinations (e.g., masking young children, self-isolation of positive cases, and restricting travel), may be beneficial in many ways [9]. However, these restrictions do not come without their problems [8, 10–12], which may jeopardize the shift into the post-acute stage of managing the pandemic; the restoration of social and economic life (World Health Organization; WHO, 2022). This study does not seek to conduct the much-needed cost/benefit analysis of restrictions nor debate public policy. Instead, it aims to identify one factor that might challenge the impartiality of such analyses: people's risk perceptions of COVID-19.

Whether to continue or re-implement restrictions is not a decision that can be made exclusively by health scientists because societal well-being cannot be reduced to a single indicator of success (i.e., reduction in deaths of COVID-19 or hospitalizations); instead, it involves consideration of competing priorities and limited resources (mental health, economics, and education, among others) [12, 13]. Therefore, this decision becomes a matter of values, where the interlinking systems of politicians, organizational decision-makers, and their constituencies direct which risks and collateral costs they are willing to accept once vaccines are shown to be effective at preventing the worst outcomes.

This study builds on recent findings suggesting that people are often misinformed when it comes to COVID-19. Their lack of knowledge, however, does not mean they do not influence others. For instance, under-estimation of the threat of COVID-19 or belief in conspiracy theories (e.g., unfounded cures) may lead people to disregard health-minded rules and risk the lives of the vulnerable [14–16]. However, being misinformed may also mean that people *over*-estimate the threat and believe COVID-19 to have far more significant negative consequences than it does [17]. Because people's tendency to over-estimate COVID-19 risks has received less attention relative to erroneous information pertaining to under-estimation, its consequences remain less understood. Nonetheless, if laypeople, who ultimately influence public policy through voting or collective action, miscalibrate the severity of risks and its mitigation tactics, their erroneous judgments may lead to depletion of scarce resources on the deleterious pursuit of poorly identified goals [4, 5, 18, 19].

Accordingly, I hypothesize that people who over-estimate COVID-19 risks will be more likely to support policies for the colloquial '*new normal*.' For this study, the *new normal* is defined as a continuation of COVID-19 risk-mitigation restrictions for an undefined period of time starting with completed vaccinations of vulnerable people (i.e., meeting the primary

global goal for the acute phase of the pandemic according to WHO). I proceed to explain how my predictions are informed by complementary theories on perceptions of unknown (vs. known) risks [19], availability and reputational cascades [4, 5, 18, 19], and moralization of COVID-19 [20, 21].

## Origins and catalysts of risk over-estimation: Theoretical foundation

In the absence of reliable metrics to gauge harm from COVID-19, a balanced risk assessment was not possible in early 2020. When a threat cannot be assessed accurately, policy-makers may embrace the *maximin principle* and choose the policy that minimizes the likelihood of a catastrophic worst-case scenario [3]. This explains why the flu, despite contributing to thousands of deaths annually [22], does not trigger restrictions; its worst-case scenarios–even with seasonal fluctuations in severity–are generally known, and its costs are tolerated. In efforts to prevent the modeled worst-case scenarios from letting COVID-19 spread, governments and health institutions sought to educate their citizens on the dangers of COVID-19, encourage health-minded behavior (e.g., physical distancing, hygiene, and masks), and combat the spread of misinformation that threatened the success of health efforts [15, 16, 23–27]. These measures have since escalated to unprecedented global travel restrictions, sealed borders, and vaccine or even booster passports to enter anywhere from schools to gyms.

While erring on the side of extreme caution is defensible in the absence of information, continuing to do so in light of new information is not. This fear-based focus can further perpetuate *availability cascades* where new information is not used to revisit the cost-benefit analysis but is selectively disseminated and censored; information that deviates from the narrative may elicit reputational damages or moral outrage [20, 21, 28]. These forces may also be explained with *moralization*, a process by which an attitude becomes a matter of moral imperative [20]. When an attitude becomes moralized, it becomes absolute, intolerant, and resistant to change, further perpetuating the availability cascade of moralized information.

Consider, for instance, contrarian opinions within the scientific community. Unlike many policy risks where public opinions tend to clash with those of the experts, COVID-19 yields diverging views and disagreements between experts [29]. Their debates are not driven by questions such as whether COVID-19 is real or how severe it is to different demographic groups, but whether the extreme and unprecedented measures applied uniformly for vulnerable and non-vulnerable alike and for extended periods are worth the cost [9, 30–33]. Yet, moralizing COVID-19 means that questioning the magnitude of risks can inflict reputational harms [4]. For example, after Dr. Ludvigsson, a pediatrician and epidemiologist, pointed out that the risks of COVID-19 to children are extremely low [34], his claims were challenged not only on empirical grounds, but he also received intimidation and personal attacks that ultimately led to him to abandon researching and debating COVID-19 [35].

Laypeople may similarly dismiss researchers and findings that go against the moralization of COVID-19. For instance, researchers [21] gave participants in New Zealand [NZ] two identical research proposals to investigate human suffering related to COVID-19. Proposals differed in one way: one wanted to examine human costs that result from abandoning elimination in NZ, and the other wanted to examine costs from continuing it. Despite containing identical information about the methodology, the proposal that challenged elimination was seen as less methodologically sound, less reliant on accurate information, and participants showed less trust in the researchers.

This phenomenon emerges in part due to the *availability heuristic* [19], a mental shortcut where the perceived likelihood of any event is dependent on how easily this event can be brought to mind. For example, information about the numbers of deaths and cases, the

dangers of long COVID-19, or overwhelmed healthcare systems is readily available. In contrast, information about the age or comorbidities of people who died [36], long-term and severe consequences of flu [37, 38], or pre-COVID-19 reports of hospitals being described as 'war zones' and needing ice truck morgues to deal with the surge in flu cases [37, 39–42] may not be cognitively available. In fact, media reports tend to focus on negative information such as deaths, hospitalizations, and cases, while giving less coverage to positive developments, such as vaccine trials in 2020 or school re-openings [43, 44]. Brookings Institute reported that Americans vastly overestimate the severity of COVID-19 by many degrees of magnitude. For example, 35% of US adults believe that half or more infected people require hospitalization for COVID-19. However, that number is likely no greater than 5% [17].

This literature suggests that left unchecked, fear-based availability cascades can perpetuate the over-estimation of risks, censorship, and eventually, rigid support and implementation of practices that disproportionately deal with one threat while undervaluing others.

### Present study

In this study, I tested the possibility that people will likely over-estimate COVID-19 risk and that their over-estimations would be associated with stronger support for continuing restrictions even after the most vulnerable populations have been vaccinated and after the threat of overwhelmed hospitals has abated. My expectations that people would over-estimate COVID-19 risks are based on scholarly evidence that media focuses [in part due to people's demand] on negative aspects of COVID-19 [43, 44] and on representative sample polling showing that people over-estimate COVID-19 risks [17]. My expectation that these over-estimates would be related to endorsement of the *new normal* is based on the literature of risk-estimation, availability heuristics, and moralization summarized above. I test a single relationship in this study: risk (anticipated over-)estimation and support for the *new normal*. I sought to increase confidence in my findings by drawing from different samples and measuring laypeople's support for the *new normal* and risk perceptions through multiple complementary indicators. In addition, I examine other variables that may inform future research on this topic [political ideology, belief in conspiracy theories that devalue COVID-19, general concern over COVID-19, and general compliance with restrictions].

## Materials and methods

All anonymized data and syntax with basic instructions are available here:
https://osf.io/87dzs/?view_only=ef9c8daf19c74e078c1c2d2abd3a06e0.

I present the results of the central hypothesis [risk over-estimation and *new normal* support] tested with core percentage-based and select alternative indicators. I rely on the Supporting Information (SI) section to present additional or exploratory findings (S1 File) that are not necessarily of central relevance to this manuscript but are nonetheless beneficial for a more holistic understanding of the data at this point (e.g., ANOVAs showing differences between samples, distributions of risk estimation variables, and sample-level results). References to SI information are noted where appropriate.

### Participants

I used diverse samples, including Mturk (Samples A and E), Prolific (Samples B and C), and community members in Australia and New Zealand (ANZ; Sample D) who were recruited to participate in the study via social media. Three hundred platform users from Mturk and Prolific were recruited for each Sample A, B, C, and E. Australian/New Zealand community members were recruited through social networks and social media ads. Those community members

were incentivized to participate such that for every complete response, $1 would be donated to one of the two charities of their choice. There was no deception, and participants' requests were honored. The community study stopped once its costs reached the available budget. Data from samples A and B were collected first (early April 2021). Data from Samples C and D were collected in mid-late April. Sample E respondents were invited to participate in early January 2022. All responses were anonymous.

The study was reviewed and approved by the University of Otago Human Ethics Committee Reference#D20-088. All participants received the information sheet outlining their rights as participants. They provided their informed consent before they began their study by selecting 'I agree to participate in this study'. If they did not wish to participate, they could choose 'I do not agree to participate' and exit the study.

I sought to increase quality by preventing participants from taking the survey more than once and introducing two attention checks embedded within perceived scientific consensus and knowledge items (sample item is "For quality control, please select '3'). The results and conclusions remained substantively unchanged regardless of whether the data were analyzed with all responses (Table 1 N Recruited) or without responses who failed the check questions (Table 1 N Retained; reported in this manuscript). Missing data were handled with pairwise deletion. Table 1 presents all sample characteristics.

## Procedures: Maximizing risk indicators and minimizing participant fatigue

Maximizing confidence in my findings required multiple complementary indicators because of the imperfections inherent in any single operationalization of COVID-19 risk perceptions among laypeople. Therefore, the study relied on: 1) *core risk indicators* or numeric estimation items that align with public surveys on COVID-19 (17) and reference information that laypeople can understand when estimating other, non-Covid risks in life (e.g., chances of surviving a cancer diagnosis), and 2) *alternative indicators* (a series of evaluative questions about COVID-19 risks). I employed a similar approach to extend the generalizability of *new normal* endorsement by measuring it as a 9-item policy scale [*new normal policy* endorsement; NNP; given to half of A-B samples, and everyone in C, D, and E), or a 3-item affect-based scale (RN-Fear; Half of A-B samples). Sub-sets of alternative indicators and items were assigned to participants at random with the goals of minimizing participant fatigue and maximizing the number of ways participants can evaluate COVID-19 risks. Varying *N*s reflect these differences. Table 2 summarizes all available study materials and content by each sample. Due to the richness of the data, the manuscript focuses on the results involving DVs, core indicators, and evaluative items (perceived scientific consensus and knowledge of risks). For other information, I direct readers to S1 File or the open data.

**Table 1. Participant characteristics by sample.**

| Sample | *N* Recruited | *N* Retained | % Male | % Female | Age *Mean* | Age *SD* | Time | Recruitment and Location |
|---|---|---|---|---|---|---|---|---|
| A | 300 | 275 | 50.20 | 47.30 | 41.2 | 13.3 | Early April '21 | Mturk; US |
| B | 300 | 294 | 65.00 | 34.00 | 29.2 | 10.5 | Early April '21 | Prolific Academic; International |
| C | 300 | 254 | 66.90 | 30.30 | 28.6 | 10.4 | Mid - late April '21 | Prolific Academic; International |
| D | 446 | 410 | 23.80 | 68.00 | 51.5 | 15.8 | Mid - late April '21 | Social Media and Community Recruitment: Australia and New Zealand |
| E | 300 | 264 | 41.80 | 55.10 | 40.5 | 12.7 | Early January '22 | Mturk; US |

*Notes*. Not all participants chose to provide their gender. They also had an option to select 'gender-diverse' (*N* = 32); on OSF their responses were merged with those who did not provide information on gender (in total *N* = 42).

**Table 2. Available study materials by sample.**

| Study Materials and Content by Sample | A<br>*Mturk*<br>*April '21* | B<br>*Prolific*<br>*April '21* | C<br>*Prolific*<br>*April '21* | D<br>*Aus/NZ*<br>*April '21* | E<br>*Mturk*<br>*January '22* |
|---|---|---|---|---|---|
| *Core DVs: New Normal Policy (NNP) Endorsement* | | | | | |
| NNP endorsement (9 items) | Half | Half | All | All | All |
| NNP: Fear of returning to *normal* (3 items) | Half | Half | - | - | - |
| *Core COVID-19 Risk Assessment Estimation Items* | All | All | All | All | All |
| 7 numerical risk indicators | | | | | |
| *Alternative COVID-19 Risk Assessment Indicators* | | | | | |
| Perceptions of scientific evidence on risk (PSE; 10 items and SI) | Half | Half | Half | Half | All |
| C19 general risk knowledge (18 items) | Half | Half | Half | Half | - |
| Long Covid' risk (6 items) | RA | RA | RA | RA | - |
| C19 outcomes based on 1,000 (vs. 100; 1 item; SI) | RA | RA | RA | RA | - |
| Perceptions of global death toll (1 item; SI) | RA | RA | RA | RA | - |
| *Current Behavior* | | | | | |
| Participation in contact-tracing | All | All | All | All | - |
| Compliance with C19 mandates | All | All | All | All | All |
| Intent to get C19 vaccine or vaccination status | All | All | All | All | All |
| *Individual Characteristics and Potential Controls* | | | | | |
| Gender | All | All | All | All | All |
| Age | All | All | All | All | All |
| Political ideology | All | All | All | All | All |
| Statistical literacy (3 items) | All | All | All | All | (1 item) |
| Belief in conspiracy theories (limited) | All | All | All | All | All |
| Personal concern over contracting C19 | -- | -- | All | All | All |
| *Exploratory Items (SI)* | | | | | |
| Right to determine cost-benefit analysis | All | All | All | All | - |
| (Health scientists, non-health scientists, public) | | | | | |
| Moral elevation in response to restrictions | -- | -- | -- | All | - |

*Notes.* RA = random assignment. *Half* = participants saw perceptions of scientific consensus or knowledge about risks questions. Items varied; the same knowledge and long Covid items were given to Samples A and B, and another set was given to participants in Samples C and D. Participants in Sample E only saw PSE items. The purpose of alternative indicators was to complement the numerical findings, and the purpose of randomly assigning only a select sub-group was to minimize participant burden. The presentation of DVs, core indicators, and alternative indicators was fully randomized and presented first; questions about behavior, individual characteristics, and exploratory items were presented last. *SI* = Supporting Information (S1 File).

## Measures

**New Normal Policy (NNP) endorsement.** Recall from the introduction that *new normal* was defined as restrictions past vaccinations. Accordingly, the primary measure of interest was examining the extent to which people support new normal policies in perpetuity. This set of items was assigned at random to half of the participants in Samples A and B, and everybody in Samples C, D, and E (see Table 2). I selected nine items that represent the most contentious and most frequently discussed issues pertaining to COVID-19 (e.g., vaccine passports, continuing contact tracing, or lifting all mandates). Participants read the following prompt question:

"Many countries have vaccination programs that are well under way. What policies should be implemented or continued once all the vulnerable people have been vaccinated and once everybody had a chance to get their vaccine? Indicate the extent to which you would

support the following policies; 1 (*I would NOT support this*) to 7 (*I would DEFINITELY support this*)"

Table 3 shows the items, means, and Cronbach's α coefficients for each sample. When examining all responses simultaneously (without sample-based divide), the scale had high internal consistency (Cronbach' α = .90) and single-factor structure; I assessed the dimensionality of the nine items with the principal axis factoring using oblique rotation, which yielded a single factor accounting for 51.4% of the item variance. Therefore, I collapsed the items to form a single score of *NNP* endorsement.

**Fear of abandoning COVID-19 restrictions (Samples A and B only).** As an affect-based complement to NNP (a policy-based assessment noted above), participants indicated how they would feel if "the world returned to 'normal' once all the vulnerable groups have been vaccinated" by selecting the extent of their agreement (1 = *strongly disagree*; 7 = *strongly agree*) with sentiments: 1) *vulnerable*, 2) *unsafe*, or 3) *worried*. Due to their high internal consistency (Cronbach's α coefficients were .96 and .92 for Samples A and B, respectively), items were collapsed to form a single scale labeled *RN-fear of returning to normal (RN-Fear)*. Because it was of secondary interest, this DV was assigned at random to half of the participants in Samples A and B only.

**Core COVID-19 estimations (numerical; all participants).** Participants estimated seven COVID-19 outcomes for people living in Western countries to attenuate the possibility that their responses would be driven by global differences in health capabilities. Table 4 shows indicators, conservative estimation benchmarks, and references for those benchmarks. The purpose of estimation benchmarks is to determine whether participants' responses significantly differ from those indicators. All participants received these questions with one exception noted below: Sample E (2022) participants evaluated Items 5, 6, and 7 with healthy, fit, *unvaccinated* individuals as the reference. All percentage-based answers were presented on a sliding scale with three anchor labels (0 = *extremely low; less than 1%*; 50 = *about 50% or half*; 100 = *extremely high; almost everybody*).

**Alternative risk indicator #2: Perceptions of Scientific Evidence (PSE) on risks and risk-mitigation claims.** Participants in Samples A–D evaluated ten COVID-19 risk-mitigation practices, and participants in Sample E (2022) evaluated 12 different items to broaden the scope of the assessment and reflect the changes in available knowledge.

**Table 3. NNP support item-based descriptive statistics and reliabilities (all samples).**

| Items | Mean | SD | N |
|---|---|---|---|
| If cases are rising, legally require to wear a face mask or covering on all public transport and flights. | **5.24** | 2.12 | *943* |
| Legally require COVID-19 vaccine passport to travel internationally. | **4.94** | 2.28 | *941* |
| Legally require COVID-19 vaccine passports to access institutions (schools, universities, recreation facilities, or workplaces). | **4.04** | 2.26 | *944* |
| If cases are rising, legally require people to wear masks ANY time they are outside (including when they are driving by themselves). | **3.93** | 2.28 | *945* |
| Implement a program similar to COVID-19 alert or levels system to contain flu, which kills thousands of people every year. | **4.13** | 2.07 | *942* |
| Continue strict contact tracing of all positive COVID-19 cases. | **5.49** | 1.96 | *942* |
| Try to eliminate COVID-19. | **5.39** | 1.95 | *942* |
| Require people who test positive for COVID-19 to self-isolate. | **6.15** | 1.57 | *938* |
| Lift ALL mandates and permit life as normal, even if there are rising cases (Reversed). | **5.48** | 2.03 | *945* |

*Note*. See S1 Table for descriptive statistics per sample.

**Table 4. Core indicators, estimation benchmarks, and references.**

| Estimations | Estimation Benchmarks | References |
|---|---|---|
| What is the average age of a person who died of COVID-19 in the (US)? | 78+ | The average age for deceased COVID-19 patients tends to be approximately 80 years of age, with some variation based on gender and reporting protocols [36, 45–51] |
| % of C19 deaths who were children | < 1% | This estimate should be below 1% [45, 50, 52, 53]. |
| % of C19 deaths who were healthy people between 18–65 | < 1% | Deaths among people < 65 without underlying health conditions remain rare [36, 47, 48, 50, 54, 55]. |
| What percent of people who get COVID-19 do NOT require ANY medical attention? That is, what % of those people recover fully without any medical treatment? | > 90%* | Public health and education outlets suggest that 'most people who contract COVID-19 recover' without any medical treatment [56]. Gudbjartsson, Norddahl [54] found that 95% of people who contract COVID-19 recover without medical treatment. Brookings Institute summarizes that "while the percentage of people who have been infected by the coronavirus needed to be hospitalized is not precisely known, that estimates varies between 1% and 5% and it is unlikely to be much higher or lower". However, this should be interpreted with caution as "An accurate calculation of infection fatality risk requires an accurate estimate of the number of infections, both diagnosed and undiagnosed" [54]. |
| End up in ICU | < 5%* | The majority of ICU hospitalizations are driven by the elderly and people with pre-existing conditions [36, 57–60]. In general, 1–5% of people who contract Covid may end up hospitalized. For a generally healthy and fit individual, those estimates should be lower. |
| Die | < 1% | The overall COVID-19 fatality rate tends to be below 1% [47] but is influenced by geographic location [61]. This estimate could also be influenced by individuals' estimation of case or infection fatality rates, which tend to differ [55] and are heavily influenced by testing and accurate diagnoses. |
| Never fully recovers from 'Long Covid' | NA | The long-term effects of COVID-19 are increasingly well-documented [56, 62–67]. At this point, no data suggests that otherwise healthy individuals might *never* fully recover from COVID-19. |

* Estimates are conservative benchmarks used to compare participants' responses against the currently available data. Therefore, these indicators should not be used as definitive indicators of COVID-19 risks. Instead, readers are directed to sources that contain far more nuanced information. **Prompt for items 1–2: "**Consider everybody who died of COVID-19 in Western countries (e.g., NZ, UK, US, EU, Australia, or Canada). What percentage (%) of those people who died of COVID-19 were". **Prompt for items 5–7:** If a generally healthy, fit person contracts COVID-19, what are the chances (%) that they will (experience one of the following outcomes). Sample E participants responded to those same questions, but the focus was on the "health, fit, unvaccinated."

All participants indicated the extent to which they believe that those practices are supported by scientific evidence (S1 File contains additional PSE items from Samples A and B). Items were selected based on the extent of their media coverage. Participants were asked to:

"Consider the amount of scientific evidence available to support each of the statements below. Select lower numbers if you think there is no or little evidence that supports that claim. Select higher numbers if you think there is clear evidence that supports that claim." 0 = *No evidence* to 6 = *Clear evidence*. Middle numbers were labeled as '*Mixed evidence*'. Numbers were recoded to a 1–7 scale.

The objective of this segment was not to quantify the *actual* scientific consensus; evidence of such consensus and occasional lack of it is evident in other sources [7, 9, 32]. Instead, the focus was on laypeople's *perceptions* of scientific consensus (PSE) and their relationship with NNP endorsement. Accordingly, I looked at practices for which there is less debate and high consensus (e.g., the efficacy of vaccines), some debate and mixed consensus (e.g., mask-wearing), and practices that are not or cannot be supported with clear consensus, such as the belief that benefits of lockdowns outweigh the risks of COVID-19, elimination (Zero-Covid) being the best global strategy to combat COVID-19, the need for wearing masks while driving or hiking alone, and closing parks and beaches. I reasoned that the evidence of risk over-estimation-NNP relationship would be stronger if participants who endorse NNPs report that claims that over-inflate risks are supported with evidence (even when they are not).

Ensuring accuracy in presentation, interpretation, and analysis pertaining to PSE items is challenging because the knowledge is evolving. I sought to reduce (and admittedly not eliminate) those challenges by employing several tactics. First, for supported claims, I relied on statements from science communication channels, and I provided appropriate references. Second, I labeled items as *supported*, *unsupported or unclear* based on currently available knowledge. Third, to aid in interpretation, I collapsed items into scales when they yielded themselves to such practice. Finally, I improved Samples A-D PSE administration and repeated the process in Sample E (2022) using more pointed questions.

Table 5 summarizes items administered to Samples A–D.

From a theoretical perspective, there are differences in items that are supported with evidence and those that are not. For instance, scoring high on item 1 is warranted (and scoring low is not), as there is strong evidence that vaccinations reduce serious illnesses, hospitalizations, and deaths from COVID-19. However, scoring high on item 4 may indicate risk over-estimation, as CDC does not suggest that people need to wear masks when enjoying outdoor activities with members of their household [90].

While there is no theoretical foundation to justify collapsing all items into a single scale, a factor analysis suggested unexpectedly that these items can be collapsed and interpreted meaningfully. Specifically, an exploratory maximum likelihood analysis [91–93] yielded a single factor accounting for 51.77% of the item variance after removing the last item (*sanitizing groceries*). Only after observing these results (i.e., I did not initially anticipate conducting these analyses), I considered evaluating the data using a 5-item scale aggregating items that have less empirical support (PSE–supported; $\alpha$ = .78), and a 5-item scale aggregating items that have more empirical support (PSE–unsupported; $\alpha$ = .86). While both yielded a single-factor structure, they were highly correlated with each other; $r$ = .70, $p < .001$, $N$ = 627. Therefore, I settled on using a 10-item aggregate when reporting correlations and regression results for Samples A–D.

Mindful of the challenges in the assessment of laypeople's perceptions of scientific evidence about COVID-19 risks, I devised a new assessment of PSE for 2022-based Sample E; one that more defensibly differentiates between: 1) six claims about risks that are *unsupported by evidence*, indicating that they are either false or unknowable, and 2) six claims about risks that are *supported* by the current evidence. Unsupported claims formed a reliable scale (Cronbach's $\alpha$ = .80) and yielded a single-factor structure accounting for 50.11% of the variance. Therefore, I collapsed these six items into a single scale to aid analyses and interpretation.

**Table 5. Perceived scientific evidence of COVID-19 risks items and correlations with NNP (Samples A–D; 2021).**

| Exact Wording | M | SD | R with NNP | N | Reference |
|---|---|---|---|---|---|
| Getting a COVID-19 vaccine helps keep you from getting seriously ill, even if you do get COVID-19. | 5.70 | 1.72 | .60** | 406 | Supported. CDC-issued statement; high consensus. [68] |
| People without COVID-19 symptoms should wear masks to minimize the spread of COVID-19. | 5.54 | 1.95 | .74** | 395 | Time-dependent guidance and mixed evidence. CDC advises mask-wearing regardless of one's symptoms or vaccination status, but their guidance fluctuates as new information becomes available [24, 69, 70]. However, this answer may be influenced by the prevalence of asymptomatic transmission [70–74]. |
| Masks are effective in protecting people against COVID-19. | 5.57 | 1.75 | .73** | 493 | CDC-issued guidelines; consensus with some mixed evidence [23, 69, 70, 75–77]. |
| The benefits of lock-downs outweigh the costs of failing to contain COVID-19. | 5.20 | 1.95 | .67** | 495 | No clear consensus. |
| In case of community outbreaks, people should wear masks ANY time they are outside, even if they are by themselves (e.g., driving or hiking). | 3.87 | 2.13 | .60** | 494 | No clear consensus [78]. While CDC and WHO recommend wearing masks outside when one cannot physically distance from others, they do not offer official rules on wearing masks while alone. |
| When a person is reported as COVID-19 death, it is clear that COVID-19 was the main cause of death. | 4.50 | 1.89 | .63** | 397 | No clear consensus at this time [79, 80]. CDC provides statistics with co-morbidities. In May 2021: "For over 5% of these deaths, COVID-19 was the only cause mentioned on the death certificate. For deaths with conditions or causes in addition to COVID-19, on average, there were 4.0 additional conditions or causes per death."[81]. Furthermore, some countries report COVID-19 deaths for people who have died within 28 days of a positive COVID-19 test [82]. |
| New variants spread faster AND are also far deadlier than the original variant. | 5.06 | 1.84 | .63** | 492 | No clear consensus [83–89] |
| In case of community outbreaks, outdoor spaces (beaches or parks) should be closed. | 4.45 | 2.08 | .57** | 403 | No clear consensus, no guidelines [90]. |
| Elimination (Zero-Covid) is the best strategy. | 4.68 | 2.07 | .43** | 495 | According to a recent *Nature* poll of COVID-19 immunologists, infectious-disease researchers, and virologists, 89% of them think that COVID-19 will become endemic and will continue to circulate around the globe, and 51% believe that elimination, even from certain regions, is unlikely [7]. More than one-third of the respondents thought that it would be possible to eliminate COVID-19 from some regions. Whether elimination is the best strategy or not remains a matter of scientific debate. |
| People should disinfect their groceries to reduce their chances of contracting COVID-19. | 3.75 | 1.95 | .74** | 395 | No CDC or WHO guidelines. |

*Note.* The ten items above were given to all Sample A–D participants. S1 File presents additional PSE items presented to each sample independently.

Assessing the factor structure of the generally supported claims, however, did not suggest that items can be collapsed into a meaningful scale even with potential recoding to infer the same direction of high/low risk, so I proceeded to treat these items as exploratory indicators and present their zero-order correlation with NNP support below.

**Alternative risk indicator 3: Knowledge-based questions about COVID-19 risks and risk mitigation (Samples A–D).** Instead of asking participants whether each risk and risk-mitigation claim had scientific support (PSE indicator noted above), participants evaluated the extent to which each item is definitely true/accurate or definitely false/misinformation (exact anchors are noted below). Recall that the purpose of alternative indicators and random administration to different samples was to increase the confidence in findings through observing the consistent relationship between risk-estimators and NNP support using different measures and anchors. Like with PSE items, I reasoned that the evidence of risk over-estimation and NNP relationship would be stronger if participants who endorse NNPs report that claims that over-inflate risks are true (even when they are not).

The participants were instructed to do the following:

"COVID-19 research continues to advance rapidly. As you are answering the following questions, consider information that is known right now and that is available through legitimate sources (e.g., the WHO, CDC, or The Ministry of Health). Some of the following statements may be true, and some may be false; others may not have a definitive answer at this time. Please indicate the extent to which each statement is TRUE or FALSE."

A sub-set of knowledge-based items focused on 'long Covid', given its potential for long-term damage [56, 62, 66]. If people over-estimate the risks of long Covid, as evident by believing or disbelieving certain claims, their over-estimation should be associated with greater support of NPPs. Instructions read:

"The following questions are about 'long Covid'—a term that describes the effects of COVID-19 that continue for weeks or months beyond the initial illness. Consider whether each of the following statements about long Covid is TRUE or FALSE."

Answer modes differed between two waves (Samples A and B), and Samples C and D. Samples A and B contained a mid-point, while those answers in Samples C and D did not. The objective was to ensure that the presence or absence of a mid-point did not influence the general patterns of the relationships [94].

**Answer mode in Samples A and B.** 1 (visually presented as *-3 = NOT true; Misinformation*) to +3 (*True; Accurate information*), with a mid-point of 0 = *partially true*. Results were recoded to a 1–7 scale.

**Answer mode in Samples C and D. 1** (*Definitely FALSE*) to 6 (*Definitely TRUE*), without a mid-point. Results were recoded to a 1–6 scale.

Tables 6 and 7 (Results) show items per Samples A/B and C/D. Sample E did not receive these items due to analytical challenges observed in Samples A–D (further noted in Discussion). Because the factor structure of 18 items did not suggest they can be meaningfully aggregated or reduced to any sub-factors, I examined their relationships on an itemized basis (Discussion highlights the challenges of doing so).

**COVID-19 current behavior (contact-tracing, compliance, and vaccine intent/status).** In Samples A–D, participants responded to three questions indicating their current COVID-19 mitigating behavior: 1) How often do you record your visits for contact tracing (e.g., manually or using a tracer app)? (1 = *almost never*; 7 = *almost every time*); 2) Overall, I have been complying with COVID-19 mandates (e.g., masks); (1 = *strongly disagree*; 7 = *strongly agree*); 3) Will you get the COVID-19 vaccine once you are eligible? (1 = *Definitely not*; 5 = *definitely yes/received it already*). Contact tracing could be influenced by one's geographical location, and it might not be a telling indicator of compliance outside of Sample C.

Participants in Sample E also reported their vaccination status because the vaccine is now more widely available (vs. A-D samples in mid-2021): 1 = *yes* (N = 205), 0 = *no* (N = 52), not coded = *I do not wish to answer* (N = 6), and indicated whether they had COVID-19: 1 = *yes* (N = 58), 0 = *no* (N = 198), not coded = *I do not wish to answer* (N = 4). They reported their general compliance with the single item: "Overall, I have been complying with COVID-19 mandates (e.g., masks)"; (1 = *strongly disagree*; 7 = *strongly agree*).

**Individual characteristics and potential controls (all participants).** The study considered several other variables that could further illuminate the nature of the results (concern over contracting COVID-19) or provide an alternative explanation of the underlying

**Table 6. Perceived scientific evidence of COVID-19 risks items and correlations with NNP (Sample E; 2022).**

| Items | N | M | SD | R with NNP | Reference |
|---|---|---|---|---|---|
| *Generally Unsupported Claims** (Aggregated)* | | | | | |
| N95 masks are safe for children use during a typical school day. | 262 | **5.15** | 1.88 | .483** | Unsupported. NIOSH-approved respirators (such as N95s) have not been tested for broad use in children [95]. |
| If US had high vaccination rates (95% or higher), COVID-19 would have stopped spreading. | 261 | **4.34** | 1.84 | .531** | Unclear. Places with high vaccination rates (Israel, Gibraltar) are experiencing surges at this time (December/January; 2022). |
| Going forward, children under the age of 12 should be prioritized for booster shots. | 263 | **3.76** | 1.85 | .507** | Unsupported. According to WHO: "While some countries may recommend booster doses of vaccine, the immediate priority for the world is accelerating access to the primary vaccination, particularly for groups at greater risk of developing severe disease [96]." |
| According to CDC, if a person is reported as COVID-19 death, COVID-19 was the CLEAR cause of death (i.e., they would still be alive if it were not for Covid). | 263 | **4.46** | 1.89 | .526** | Unclear. Not obvious, as the average person who died of COVID-19 has had 2+ major comorbidities [59]. |
| The benefits of lock-downs outweigh the risks of failing to contain COVID-19. | 263 | **4.53** | 1.98 | .610** | Unknowable [13, 97]. |
| To reduce the risk of COVID-19 infection, people should wear a mask even when they can socially distance (e.g., while hiking, exercising, or driving alone). | 263 | **4.00** | 2.24 | .469** | Unsupported. Wearing masks outside while being able to socially distance is not necessary, according to CDC [90]. |
| *Generally Supported Claims (Not aggregated)* | N | M | SD | R with NNP | Reference |
| Most hospitalizations (with Covid) at this time are comprised of unvaccinated people. | 262 | **5.43** | 1.68 | .551** | Supported but dependent on geographical location [98]. |
| Vaccines reduce the risk of hospitalizations and deaths. | 262 | **5.98** | 1.52 | .503** | Supported [99]. |
| Natural immunity is effective against a reinfection. | 263 | **3.88** | 1.79 | -.400** | Supported [7, 100–103]. |
| Most people will get Covid. | 263 | **4.91** | 1.63 | -0.091 | Supported; *Nature* survey suggests that most epidemiologists believe that COVID-19 will become endemic [7, 104]. |
| Boosters do not stop transmission of Omicron variant. | 262 | **4.66** | 1.81 | -.291** | Supported; Boosters appear effective at reducing hospitalizations and deaths, and possibly reducing transmissions, but there is no evidence that they *stop* transmission [104, 105]. |
| Most people with COVID-19 recover completely and return to normal health. | 263 | **5.02** | 1.63 | -.346** | Supported [106, 107]. |

relationship: 1) political ideology, 2) basic statistics literacy, and 3) COVID-19 denialism and conspiracy beliefs. I selected those latter three variables as potential controls because liberals tend to be more concerned with COVID-19 [21], and they tend to be more likely to over-estimate its risks [17], people who have greater statistics literacy tend to make more accurate COVID-19 risk assessments [108], and people who believe conspiracy theories tend to have warped perceptions of COVID-19 risks [23, 26, 109].

*Concern over contracting COVID-19 (Samples C, D, E].* I anticipated that greater COVID-19 risk-estimation would be related to greater concern over contracting COVID-19 (I did not hypothesize the directionality of this relationship; I explain why at length in the Discussion). Participants were asked: "How concerned or worried would you be if you or somebody close to you got COVID-19?" and provided their answer on a slider scale from 0 (*not at all concerned*) to 100 (*extremely concerned*).

*Political ideology / conservatism (all participants).* Participants indicated their political ideology on a scale from 1 = *very liberal or left-wing* to 7 = *very conservative or right-wing*. Participants in Sample E also noted who they voted for (1 = *Biden*; 0 = *Trump*, missing = *other, do not wish to answer*).

*Statistics literacy (Limited administration).* Participants in Samples A–D were asked to solve three basic statistics problems.

**Table 7. Descriptive statistics: Estimation of negative COVID-19 consequences per sample.**

| Estimations | Estimation Benchmarks | Sample A: Mturk | | | Sample B: Prolific | | | Sample C: Prolific | | | Sample D: NZ Community | | | Sample E: Mturk '22 | | |
|---|---|---|---|---|---|---|---|---|---|---|---|---|---|---|---|---|
| | | *Mean* | *Median* | *SD* | *Mean* | *Median* | *SD* | *Mean* | *Median* | *SD* | *Mean* | *Median* | *SD* | *Mean* | *Median* | *SD* |
| What is the average age of a person who died with Covid-19? | 78+ | **63.97** | 65.00 | 11.84 | **66.97** | 70.00 | 11.76 | **64.32** | 65.00 | 11.91 | **66.11** | 70.00 | 12.56 | **60.42** | 65.00 | 12.61 |
| % of C19 deaths who were children | < 1% | **6.69** | 4.00 | 7.60 | **8.73** | 5.00 | 9.66 | **9.96** | 5.00 | 11.80 | **9.13** | 6.00 | 10.85 | **11.64** | 7.00 | 14.03 |
| % of C19 deaths who were healthy people between 18–65 | < 1% | **30.88** | 24.00 | 25.51 | **35.35** | 28.00 | 27.70 | **40.19** | 33.00 | 28.51 | **30.33** | 25.00 | 24.08 | **36.36** | 30.00 | 26.12 |
| % of people who recover without medical intervention | > 90% | **67.22** | 75.00 | 26.69 | **60.01** | 70.00 | 28.06 | **62.66** | 69.00 | 22.67 | **67.07** | 73.00 | 24.53 | **67.54** | 75.00 | 25.31 |
| % that a healthy person < 65 ends up in ICU | < 5% | **19.14** | 11.00 | 18.93 | **24.35** | 19.00 | 19.99 | **17.08** | 11.00 | 16.58 | **15.59** | 10.00 | 16.64 | **26.11** | 15.00 | 26.24 |
| % that a healthy person < 65 dies | < 1% | **10.46** | 5.00 | 15.23 | **14.15** | 6.00 | 18.64 | **9.11** | 4.00 | 13.88 | **9.40** | 3.00 | 14.80 | **16.54** | 5.00 | 22.66 |
| % that a healthy person < 65 never fully recovers from Long Covid | NA | **17.89** | 9.00 | 21.14 | **20.23** | 10.00 | 21.91 | **19.48** | 10.00 | 21.56 | **20.99** | 12.50 | 22.30 | **24.60** | 10.00 | 27.60 |

* Estimation benchmarks are only used for statistical purposes to determine whether participants' responses differ from this estimate. They are used as a referenced test value in a one-sample *t* test for each DV. Samples A–D are from April 2021. Sample E is from January 2022.

*Problem 2*: "Consider this fictional statement: 15% of people like surveys. This means that X people like surveys." Options were: a) 1500 out of 5, b) 0.15 out of 100, c) 15 out of 150, d) 15 out of 100, and e) 15 out of 1,000. (% correct = 97.5).

*Problem 3*: "200 people took a test. 20 of those people scored 100% on that test. Which of the following conclusions is TRUE?" Options were: a) 10% of the people got all questions right; b) 100% of the people got 20 questions right; c) 20% of people got all questions right; and d) 10% of the people failed the test. (% correct = 83.1).

*Problem 3*: "A school has 1000 students. Only 1 student walked to school in 2020. In 2021, 800% more students started walking to school. How many MORE students are walking to school in 2021 than they did in 2020?" Options were: 8, 18, 80, or 800. (% correct = 53.6).

Participants' final scores ranged from 0 = *missed all three questions* to 3 = *correctly answered all three questions*. Problem 3 emerged to be significantly correlated with COVID-19 risk estimations, so I reported its results separately for exploratory purposes, and I included it in Sample E (2022) data collection.

**Covid denialism and conspiracy support.** *Samples A–D (limited).* Belief in COVID-19 conspiracy theories was assessed with the following items: 1) COVID-19 virus is not real; it does not exist, 2) COVID-19 is caused by 5G networks, 3) COVID-19 virus is made in a lab; and 4) 'Long Covid' is not a medically-documented condition. Depending on the version of the survey, participants responded to at least two of those items. This response method was not by design. Instead, I made an error in survey coding, which resulted in participants not seeing all intended items. Items were embedded within perceived scientific consensus and general knowledge alternative indicators. Because of different anchors and the error, responses

were standardized so that 1 = *dismissal of conspiracy theory statements*, and 10 = *belief in conspiracy theory statements.*

Since the data collection (April 2021) and while the manuscript was under review, the so-called "lab leak hypothesis" (i.e., belief that COVID-19 emerged from a lab) has received more attention, and according to Maxmen and Mallapaty [110], many research institutes are calling for a deeper investigation of this possibility. Because of the imperfections of this and other items (including an error in administration), this variable in Samples A–D should be interpreted with caution.

*Sample E (2022)*. To address the shortcomings noted above, Sample E noted the extent to which the following statements are supported with evidence (1 = *no evidence*; 7 = *clear evidence*): 1) Virus that causes COVID-19 has never been isolated; 2) PCR tests cannot differentiate flu from COVID-19; 3) COVID-19 pandemic has been planned by the global powers. There is no research that would suggest any of these statements are true. Items were collapsed into a single scale (Cronbach's α = .70), indicating beliefs in conspiracy theories.

## Results

### PART I: Predicting *new normal policy* support with core risk indicators

The underlying prediction was that people would over-estimate COVID-19 risks and that this over-estimation would be associated with stronger support of continuing restrictions. However, while determining what makes the label '*over-estimation*' a more appropriate descriptor of this phenomenon as opposed to '*higher estimation*' is a matter of semantics, a statistically defensible answer requires an assessment of participants' responses against the currently available data. Therefore, the first step was to assess whether participants' responses are over-estimates (and thus erroneous) or whether they are higher estimates but still within the expected range. Table 8 shows participants' responses on core indicators. Table 4 in *Methods* lists the known risks of COVID-19 and provides relevant recent resources used to inform those estimation benchmarks.

Results of one-sample *t* tests revealed that participants' estimations were significantly different from the estimation benchmarks ($p < .0001$) and emerged in every sample independently and for every risk indicator (detailed results are provided in S4 Table).

S1 Fig provides additional data based on 2021 samples (including a visual representation of frequency distributions) which can aid readers in evaluating this work (also see S2 Table). The distribution of the responses suggests that according to the currently available COVID-19 risk statistics, the percentage of people who under-estimate COVID-19 across all four samples is small. For instance, 34 people (2.8%) estimated the average age of COVID-19 death to be above 82, 154 people (12.49%) estimated that more than 90% of the people recover, and only 63 (5%) estimated that more than 95% of the people recover without any medical intervention. Those recovery estimates are still within the possible range.

Table 9 presents zero-order correlations between the core risk indicators, NNP endorsement, and core demographic variables for Samples A–D due to their shared historical context. Table 10 presents those statistics for Sample E, because it is the only data collected in 2022. Correlations table results suggest that higher estimations of negative outcomes of COVID-19 (e.g., deaths among children), and lower estimations of positive outcomes (e.g., recovery) are consistently associated with increased desire to continue restrictions, as operationalized as both NNP support (all samples) and fear of returning to normal after vaccinations (RN-Fear administered in Samples A and B only). Estimations of negative COVID-19 outcomes are related to COVID-19 compliance behavior, with some exception in regards to contact tracing, which may indicate differences in local laws. Finally, these basic correlations results also show

**Table 8. Descriptive statistics and correlation coefficients between risk estimation (Core) and 'new normal' support (Samples A–D; 2021).**

| Variables | Mean | SD | N | 3 | 4 | 5 | 6 | 7 | 8 | 9 | 10 | 11 | 12 | 13 | 14 | 15 | 16 | 17 | 18 | 19 | 20 |
|---|---|---|---|---|---|---|---|---|---|---|---|---|---|---|---|---|---|---|---|---|---|
| 1. NNP endorsement | 4.98 | 1.55 | 947 | -.28** | .27** | .34** | -.37** | .34** | .25** | .35** | .35** | .63** | .69** | -.09** | -.04 | -.37** | -.40** | -.06 | .04 | .61** | .84** |
| 2. RN - fear | 3.47 | 1.66 | 280 | -.05 | .21** | .15* | -.20** | .26** | .26** | .32** | .17** | .29** | .24** | -.19** | .15* | -.24** | -.08 | -.11 | .03 | -- | .39** |
| 3. Average age of C19 death | 65.46 | 12.12 | 1204 | | -.26** | -.39** | .28** | -.28** | -.22** | -.25** | -.08** | -.18** | -.14** | .04 | .06* | .06 | .04 | .06* | -.04 | -.19** | -.23** |
| 4. % of C19 deaths who were children | 8.66 | 10.20 | 1233 | | | .41** | -.32** | .43** | .43** | .37** | .15** | .12** | .11** | -.12** | -.05 | -.07* | -.01 | -.13** | .09** | .17** | .26** |
| 5. % of C19 deaths (healthy; 18–65) | 33.68 | 26.49 | 1233 | | | | -.34** | .42** | .38** | .36** | .09** | .24** | .17** | -.05 | -.13** | -.06* | -.09** | -.10** | .12** | .30** | .37** |
| 6. % recover without medical intervention | 64.51 | 25.70 | 1232 | | | | | -.46** | -.37** | -.34** | -.09** | -.20** | -.20** | .04 | .04 | .08** | .11** | .12** | -.11** | -.23** | -.36** |
| 7. % that a healthy person < 65 ends up in ICU | 18.78 | 18.29 | 1233 | | | | | | .76** | .57** | .07* | .17** | .15** | -.12** | -.08** | -.05 | -.03 | -.21** | .18** | .27** | .32** |
| 8. % that a healthy person < 65 dies | 10.71 | 15.83 | 1233 | | | | | | | .61** | .08** | .13** | .08** | -.13** | -.03 | -.01 | .03 | -.20** | .15** | .16** | .23** |
| 9. %healthy < 65 never fully recovers | 19.81 | 21.80 | 1233 | | | | | | | | .17** | .21** | .22** | -.19** | .00 | -.12** | -.08** | -.13** | .09** | .32** | .32** |
| 10. Contact-tracing | 3.13 | 2.42 | 1219 | | | | | | | | | .24** | .30** | -.24** | .31** | -.21** | -.15** | .02 | -.04 | .28** | .32** |
| 11. General compliance | 6.10 | 1.51 | 1218 | | | | | | | | | | .55** | -.13** | -.04 | -.25** | -.33** | -.01 | -.01 | .48** | .58** |
| 12. Vaccine intent | 4.21 | 1.30 | 1215 | | | | | | | | | | | -.03 | -.01 | -.34** | -.39** | .06* | -.03 | .48** | .63** |
| 13. Gender (1 = male) | 0.50 | 0.50 | 1195 | | | | | | | | | | | | -.27** | .11** | .02 | .17** | -.10** | -.10* | -.04 |
| 14. Age | 39.03 | 16.36 | 1213 | | | | | | | | | | | | | .03 | .02 | -.06* | .01 | -.05 | -.06 |
| 15. Political ideology (Conservatism) | 4.07 | 2.06 | 1138 | | | | | | | | | | | | | | .27** | -.06 | .05 | -.23** | -.38** |
| 16. Conspiracy beliefs | 2.74 | 1.80 | 1229 | | | | | | | | | | | | | | | -.08** | .03 | -.24** | -.43** |
| 17. Statistical literacy | 1.88 | 1.13 | 1233 | | | | | | | | | | | | | | | | -.72** | -.06 | -.06 |
| 18. Stat. Item 3 | 0.33 | 0.47 | 1204 | | | | | | | | | | | | | | | | | .04 | .03 |
| 19. Concern | 61.59 | 35.01 | 658 | | | | | | | | | | | | | | | | | | .59** |
| 20. PSE | 4.77 | 1.38 | 627 | | | | | | | | | | | | | | | | | | |

\* = $p < .05$,

\*\* = $p < .01$; Gender (1 = *male*; 0 = *female*)

*Notes.* The results above are presented with all samples for simplicity. S5 Table presents the same tables by sample. ANOVA results presented in S3 Table show that there are mean differences in core variables between samples. Recall that the reason for focusing on multiple samples was to expand the generalizability of the core relationship (i.e., risk over-estimation and NNP). Therefore, between-group comparisons were not of interest but are documented in SI. SI (S9 Table) also presents results without outliers to show that the conclusions persist even after eliminating responses that greatly over- or under-estimate risks (e.g., indicate that the average age of a person who died of COVID-19 is 100).

Table 9. Descriptive statistics and correlation coefficients between risk estimation (Core) and 'new normal' support (Sample E; 2022).

| Variables | Mean | SD | N | 2 | 3 | 4 | 5 | 6 | 7 | 8 | 9 | 10 | 11 | 12 | 13 | 14 | 15 | 16 | 17 | 18 | 19 |
|---|---|---|---|---|---|---|---|---|---|---|---|---|---|---|---|---|---|---|---|---|---|
| 1. NNP Support | 4.64 | 1.64 | 263 | -.34** | .19** | .30** | -.38** | .42** | .40** | .39** | .64** | .38** | -.18** | -.06 | -.06 | -.51** | .57** | -.24** | -.12 | .67** | .74** |
| 2. Average age of C19 death | 60.42 | 12.61 | 260 | | -.33** | -.34** | .26** | -.35** | -.39** | -.36** | -.25** | -.15* | .10 | .03 | .14* | .12 | -.21** | -.10 | .20** | -.31** | -.35** |
| 3. % of C19 deaths who were children | 11.64 | 14.03 | 263 | | | .50 | -.29** | .54** | .51** | .49** | .03 | .00 | .02 | .01 | -.11 | .10 | .09 | .33** | -.27** | .26** | .23** |
| 4. % of C19 deaths (healthy; 18–65) | 36.36 | 26.12 | 263 | | | | -.38** | .53** | .46** | .52** | .21** | .10 | -.12 | -.16* | -.09 | -.02 | .11 | .11 | -.26** | .35** | .33** |
| 5. % recover without medical intervention | 67.54 | 25.31 | 263 | | | | | -.36** | -.32** | -.36** | -.24** | -.11 | .18** | .10 | .02 | .14* | -.12 | .07 | .30** | -.28** | -.28** |
| 6. % that a healthy person < 65 ends up in ICU | 26.11 | 26.24 | 263 | | | | | | .80** | .78** | .24** | .16* | -.09 | -.12 | .04 | -.05 | .19** | .21** | -.36** | .45** | .42** |
| 7. % that a healthy person < 65 dies | 16.54 | 22.66 | 263 | | | | | | | .75** | .26** | .14* | -.11 | -.13* | .05 | -.03 | .19** | .27** | -.27** | .41** | .36** |
| 8. %healthy < 65 never fully recovers | 24.60 | 27.60 | 263 | | | | | | | | .25** | .16* | -.18** | -.10 | .01 | -.07 | .21** | .19** | -.29** | .40** | .39** |
| 9. General compliance | 6.11 | 1.28 | 263 | | | | | | | | | .37** | -.16* | -.04 | .04 | -.35** | .44** | -.25** | .08 | .53** | .58** |
| 10. Vax. Status | 0.80 | 0.40 | 257 | | | | | | | | | | -.08 | .07 | .08 | -.28** | .36** | -.34** | -.04 | .26** | .45** |
| 11. C19 recovery status | 0.23 | 0.42 | 256 | | | | | | | | | | | .08 | -.07 | .15* | -.14* | .07 | .07 | -.18** | -.14* |
| 12. Gender (1 = male) | 0.47 | 0.59 | 259 | | | | | | | | | | | | -.07 | .07 | -.04 | .01 | .16* | -.10 | .05 |
| 13. Age | 40.49 | 12.66 | 263 | | | | | | | | | | | | | .12 | -.10 | -.11 | .01 | .01 | -.09 |
| 14. Political ideology (Conservatism) | 4.26 | 2.37 | 262 | | | | | | | | | | | | | | -.66** | .41** | -.42** | -.42** | -.45** |
| 15. Past voting (1 = Biden) | 0.70 | 0.46 | 211 | | | | | | | | | | | | | | | -.27** | .01 | .43** | .56** |
| 16. Conspiracy | 2.57 | 1.37 | 263 | | | | | | | | | | | | | | | | -.19** | -.09 | -.17 |
| 17. Stat. Item. (1 = correct) | 0.48 | 0.50 | 263 | | | | | | | | | | | | | | | | | -.18** | -.06** |
| 18. Concern | 58.78 | 32.71 | 263 | | | | | | | | | | | | | | | | | | .55 |
| 19. PSE-False | 4.70 | 1.37 | 263 | | | | | | | | | | | | | | | | | | |

**Table 10. Predicting NNP and RN-Fear with core indicators of risk (Samples A-D).**

| | NNP (DV) | | | RN-Fear (DV) | | |
|---|---|---|---|---|---|---|
| **Predictors (Samples A–D)** | **B** | **SE** | **p** | **B** | **SE** | **p** |
| Average age of C19 death | -0.01 | -0.01 | .001 | 0.01 | 0.01 | .326 |
| % of C19 deaths: Children | 0.01 | 0.01 | .283 | 0.02 | 0.01 | .188 |
| % of C19 deaths: Healthy between 18–65 | 0.01 | 0.01 | .000 | 0.00 | 0.00 | .673 |
| % recover without intervention | -0.01 | -0.01 | .000 | -0.01 | 0.00 | .196 |
| Healthy person - Chances (collapsed) | 0.02 | 0.02 | .000 | 0.03 | 0.01 | .000 |
| *F* (df) | 53.89 (5, 913) | | | 7.09 (5, 273) | | |
| | *p* < .001 | | | *p* < .001 | | |

*Note*. Individual regression analyses by the sample are shown in S6 Table.

that those who are more likely to endorse the *new normal* are people who identify as women, are liberal, are more personally concerned about contracting COVID-19, who currently report compliance with health-minded behavior, and who are less likely to believe in COVID-19 conspiracy theories.

**Regression results.** Next, I conducted regression analyses to examine the relationship between core COVID-19 indicators and NNP support, considering the interrelated nature of multiple indicators. Specifically, I: 1) assessed the effects of core indicators on NNP, 2) examined this relationship while considering the potential controls, and 3) assessed the predictive power of perceived scientific consensus on NNP over and above all other variables. Because of the high correlation between three core indicators (estimate the chances of a healthy person dying, ending up in ICU, or never recovering; $r > .70$, $p < .001$), I averaged them into a single estimate to reduce multicollinearity.

First, I assessed the predictive power of the core indicators on NNP and RN-fear. Table 11 results (based on all participants) suggest that core indicators predict NNP support, and all but one (percentage of COVID-19 deaths that are children) were significantly related to NNP. SI (S6–S8 Tables) presents sample-specific regression results; while the core indicators predict NNPs, some individual indicators are not statistically significant, which could be due to lower power.

Next, I examined whether the core indicators continue to predict NNP even after controlling for factors that can offer competing explanations. Accordingly, I entered the potential controls under Step 1 and core indicators under Step 2. Step 1 controls included gender due to

**Table 11. Predicting NNP with core indicators of risk (Sample E).**

| | NNP (DV) | | |
|---|---|---|---|
| **Predictors (Sample E)** | **B** | **SE** | **p** |
| Average age of C19 death | -0.02 | 0.01 | .002 |
| % of C19 deaths: Children | -0.02 | 0.01 | .029 |
| % of C19 deaths: Healthy between 18–65 | 0.00 | 0.00 | .631 |
| % recover without intervention | -0.02 | 0.00 | .000 |
| Healthy person - Chances (collapsed) | 0.02 | 0.01 | .000 |
| *F* (df) | 19.92 (5, 254) | | |
| | *p* < .001 | | |

*Note*. RN-Fear was not included in Sample E data collection due to its inconsistent findings from previous data collections.

its positive relationship with NNP, political ideology, and conspiracy beliefs. Although I planned on controlling for statistics knowledge, I did not do so as this variable was shown to be uncorrelated with NNP. Finally, because my previous analyses unexpectedly revealed that PSE (perceived scientific evidence of COVID-19 risks) is strongly related to NNP, I examined whether it predicts NNP more strongly than other variables by entering it in Step 3. I clarify that I made this decision only after observing the positive results (i.e., its strong relationship with NNP was unexpected and I intended to examine it separately). I present analyses by Samples A–D (Table 12) and Sample E (Table 13) separately because they used different wording and because data were collected in different years. Accordingly, Sample E also considers participants' COVID-19 recovery and vaccination status as potential controls.

Results presented in Tables 11–14 show that core risk-estimation indicators predict NNP support, and this relationship emerges even after controlling for gender (included due to its positive relationship with NNP shown in Table 10), political ideology, conspiracy beliefs, and specific to Sample E, vaccination status and history of COVID-19 recovery. In summary, core risk indicators jointly predict NNP support and RN-Fear, but the latter is only predicted by perceptions that a healthy person will suffer adverse outcomes. To be sure, not all indicators emerged as strong predictors; underestimating recovery without intervention and overestimating risks to healthy people (collapsed variable) emerged as strong predictors of NNP, but the estimation of the percent of global deaths that were children and healthy people did not emerge as consistent predictors of NNP support when assessing the data as separate samples. Indicating that COVID-19 risks and mitigation practices are supported with scientific evidence (even when they are not) predicted NNP even after controlling for other variables (entered in Steps 1 and 2).

## PART II: Exploring the relationship between NNP support and risk perception using the knowledge-based alternative indicators

This section shows participants' responses to a range of claims about COVID-19 risks and risk-mitigation tactics. Recall that these items were varied in a way that Samples A/B and Samples C/D received the same items and anchors. The intention was to categorize items that may be seen as over- or under-estimates. Unlike the perceived scientific evidence items, knowledge-based indicators did not form an interpretable factor structure. Therefore, to get a wider

**Table 12. Predicting NNP support with core indicators and controls (Samples A–D).**

| Predictors | Step 1 | | | Step 2 | | | Step 3 | | |
|---|---|---|---|---|---|---|---|---|---|
| | *B* | *SE* | *p* | *B* | *SE* | *p* | *B* | *SE* | *p* |
| Gender (1 = male) | 0.05 | 0.13 | .708 | 0.25 | 0.12 | .029 | 0.12 | 0.08 | .139 |
| Conspiracy beliefs | -0.20 | 0.03 | .000 | -0.17 | 0.03 | .000 | -0.05 | 0.02 | .010 |
| Ideology (conservatism) | -0.20 | 0.03 | .000 | -0.19 | 0.03 | .000 | -0.05 | 0.02 | .029 |
| Average age of C19 death | | | | -0.01 | 0.01 | .219 | -0.01 | 0.00 | .078 |
| % of C19 deaths: Children | | | | 0.01 | 0.01 | .260 | 0.00 | 0.00 | .500 |
| % of C19 deaths: Healthy between 18–65 | | | | 0.01 | 0.00 | .016 | 0.00 | 0.00 | .200 |
| % recover without intervention | | | | -0.01 | 0.00 | .000 | 0.00 | 0.00 | .027 |
| % outcomes for a healthy person (collapsed). | | | | 0.02 | 0.00 | .000 | 0.01 | 0.00 | .003 |
| Perception of scientific evidence | | | | | | | 0.79 | 0.04 | .000 |
| *F* (df) | 70.92 (3, 429) | | | 38.54 (8, 424) | | | 119.41 (9, 423) | | |
| | *p* < .001 | | | *p* < .001 | | | *p* < .001 | | |
| *R*² | 0.22 | | | 0.42 | | | 0.72 | | |
| *R*² Δ | | | | .20; *p* < .001 | | | .30; *p* < .001 | | |

**Table 13. Predicting NNP support with core indicators and controls (Sample E).**

| Predictors | Step 1 B | SE | p | Step 2 B | SE | p | Step 3 B | SE | p |
|---|---|---|---|---|---|---|---|---|---|
| Gender (1 = male) | -0.03 | 0.16 | .829 | 0.17 | 0.14 | .216 | -0.05 | 0.12 | .680 |
| Conspiracy beliefs | 0.05 | 0.08 | .523 | -0.12 | 0.07 | .083 | -0.08 | 0.06 | .158 |
| Ideology (conservatism) | -0.30 | 0.04 | .000 | -0.26 | 0.04 | .000 | -0.12 | 0.03 | .000 |
| Vaccination status | 1.12 | 0.24 | .000 | 0.63 | 0.21 | .003 | 0.09 | 0.18 | .637 |
| Covid recovery status | -0.38 | 0.21 | .076 | -0.11 | 0.19 | .573 | -0.07 | 0.16 | .648 |
| Average age of C19 death | | | | -0.02 | 0.01 | .003 | -0.01 | 0.01 | .124 |
| % of C19 deaths: Children | | | | 0.00 | 0.01 | .921 | 0.00 | 0.01 | .624 |
| % of C19 deaths: Healthy between 18–65 | | | | 0.00 | 0.00 | .838 | 0.00 | 0.00 | .593 |
| % recover without intervention | | | | -0.01 | 0.00 | .003 | -0.01 | 0.00 | .007 |
| % outcomes for a healthy person (collapsed). | | | | 0.02 | 0.00 | .000 | 0.01 | 0.00 | .013 |
| Perception of scientific evidence | | | | | | | 0.67 | 0.07 | .000 |
| $F$ (df) | 24.42 (5, 240) | | | 25.71 (10, 235) | | | 42.36 (11, 234) | | |
| | $p < .001$ | | | $p < .001$ | | | $p < .001$ | | |
| $R^2$ | 0.34 | | | 0.52 | | | 0.66 | | |
| $R^2 \Delta$ | | | | .19; $p < .001$ | | | .14; $p < .001$ | | |

understanding of participants' risk perceptions, I report the means and basic correlations with NNP support.

Part II results section should be interpreted with caution due to their itemized reporting. Even though these items did not yield an interpretable structure that would allow cleaner analyses, I share these findings so that others may be more successful in assessing laypeople's perceptions of risk. Despite challenges in interpreting item-based correlations, a few telling patterns emerge, which can be used to refine these questions and promote future research. For instance, NNP support was related to the perception that COVID-19 was the main cause of death in the US (2020; incorrect), labeling as incorrect the claim that UK COVID-19 deaths are reported if they occur within 28 days of a positive test, and labeling as incorrect the claim that people over 80 account for half of all COVID-19 deaths (see Methods for caveats).

## General discussion

I examined whether COVID-19 risk perceptions (and specifically over-estimations) are related to endorsement of the *new normal*; continuing restrictions such as vaccine passports, masking mandates, self-isolation, and pursuit of COVID-19 elimination after all the vulnerable groups have been vaccinated and once everybody had a chance to get the vaccine. Efforts to prevent the spread of misinformation have primarily targeted arguments that under-estimated or dismissed the threat of COVID-19, or that gave credence to questionable origin stories and untested solutions [26]. I conducted this study with the hope of not supplanting but *complementing* this predominant perspective and encouraging others to consider the possibility that uncorrected misinformation that over-estimates risks and presents information partially–particularly now—may be as damaging to social and health recovery as uncorrected misinformation that under-estimates them.

Drawing from findings based on 1,500+ participants, distinct data collections, and multiple assessments of risk, I summarize the key trends emerging from the data, highlight the limitations, challenges, and boundaries of the results, and discuss implications and need for future research on COVID-19 risk-estimation.

**Table 14. NNP and relationship with knowledge questions in Samples A and B.**

| Item | M (1–7) | SD | r with NNP | r with RN-Fear | References for expecting participants to note 'true' or 'false/misinformation'. |
|---|---|---|---|---|---|
| Sun rays can neutralize COVID-19. | 2.54 | 1.88 | -.201* | 0.15 | **True** according to Ratnesar-Shumate, Williams [111] who note: "Ninety percent of infectious virus was inactivated every 6.8 minutes in simulated saliva and every 14.3 minutes in culture media when exposed to simulated sunlight representative of the summer solstice at 40˚N latitude at sea level on a clear day." |
| In the UK, if a person dies within 28 days of a positive COVID-19 test, they are counted as COVID-19 death (even if the person was in terminal stages of another illness). | 4.32 | 1.69 | -.320** | -0.01 | **Factual statement–True** [112, 113]. |
| While accounting for a small portion of the population, people over 80 accounted for around half of all COVID-19 deaths. | 5.20 | 1.40 | -0.16 | -0.14 | **Generally true** [45]. When assessing COVID-19 risks, people generally under-estimate the risk to older adults. |
| In many Western countries, the majority of all COVID-19 deaths occurred in aged care facilities. | 4.59 | 1.54 | -0.15 | -0.15 | **Generally true** with the first wave of COVID-19 in Australia with 75% [114], Belgium with 61.3% [115], and Canada with more than 80% [116]. However, with greater awareness of aged care facilities and vaccines, this percentage is decreasing [117] |
| In 2020, COVID-19 was the main cause of death in the US. | 4.03 | 2.12 | .232** | 0.11 | **Factual statement–False.** [118]. COVID-19 was the third cause of death, with heart disease [690,882] and cancer [598,932] claiming more lives than COVID-19 [345,323]. [118]. More than 581,000 people have died of Covid since 2020, which is still lower than the number of people who died of heart disease and cancer in 2020. |
| Sun and warm weather protect people against COVID-19. | 2.51 | 1.68 | -.215* | 0.02 | **False.** [25] |
| The risk of outdoor transmission of COVID-19 is high. | 3.36 | 1.96 | .287** | 0.13 | **False.** This survey was administered before the widespread discussion of outdoor transmission took place in early May 2021 [119]. |
| The risk of surface transmission of COVID-19 is high. | 3.97 | 2.01 | 0.12 | .291** | **False.** CDC summarizes: "the risk of SARS-CoV-2 infection via the fomite transmission route is low, and generally less than 1 in 10,000, which means that each contact with a contaminated surface has less than a 1 in 10,000 chance of causing an infection" [120] |
| Children are considered a HIGH risk group. | 1.90 | 1.45 | 0.16 | .224** | **Factual statement–False** [36, 121]. |
| For children, COVID-19 is far deadlier than flu. | 3.09 | 1.89 | .211* | .205* | **Unclear.** |
| For healthy, fit people under 25, COVID-19 is no more dangerous than flu. | 3.48 | 1.81 | -.490** | -.428** | **Unclear.** This might be *generally true* if dangers were reduced to a binary variable where the outcome is either full recovery or death [117]. Support of this statement as truthful would also require an assumption that *Long Covid* complications are comparable [not worse] to those of any post-viral syndrome, including flu [37]. |
| The majority of COVID-19 spread is by people who are infected but show no symptoms (e.g., people with no fever or cough). | 5.28 | 1.59 | .231** | 0.05 | **Unclear, mixed evidence** [70, 71, 74, 122–125]. |
| (LC)*. Many documented Covid symptoms are primarily psychological (e.g., anxiety). | 2.78 | 1.64 | -.165* | -0.16 | **True.** [64, 67, 126, 127]. |
| Long Covid symptoms include psychological and neurological disorders. | 4.77 | 1.69 | .250** | .238** | **True.** [67, 127, 128]. |
| Most people recover completely within a few weeks. | 5.18 | 1.56 | -.313** | -.264** | **True.** Mayo Clinic notes: "Most people who have coronavirus disease 2019 [COVID-19] recover completely within a few weeks". Additional information: [54, 106] |
| A patient hospitalized for Covid may experience same long-haul symptoms as a patient hospitalized for any reason. | 4.32 | 1.63 | 0.01 | -0.15 | **True.** [129] |
| A third of all people who test positive for COVID-19 will have lung scarring for at least 6 months. | 3.94 | 1.72 | .377** | .289** | **False.** This statement represents a significant over-estimation of risk. There is no evidence that one-third of positive cases may have those symptoms for six months. |
| Long-term lung scarring and heart inflammation are unique to COVID-19 (i.e., not found in flu). | 4.49 | 1.88 | .196* | .172* | **False.** Flu results in lung scarring and heart inflammation [37, 38]. Heart inflammation [inflammatory disease of the myocardium; myocarditis] often results from "common viral infections and post-viral immune-mediated responses"[130]. |

*Note.* Items 12–18 pertain to 'Long Covid' (LC).

## Risk over- vs. under-estimation

When interpreting the core indicator findings, it is first essential to evaluate whether the results are driven by *higher* estimations of COVID-19 risks (i.e., estimations that are high but within the possible bounds) or whether they are driven by *over*-estimations of risk. In other words, at what threshold does the estimate become *over*-estimate? While these distinctions are matters of semantics, the results suggested that people over-estimated the negative consequences of COVID-19 (i.e., they provided lower average age of death, greater estimation of deaths and hospitalizations for children and healthy people under the age of 65), and they under-estimated the positive consequences (i.e., the potential for recovery without medical assistance). This trend emerged independently in all five samples (A–D based in 2021, and E based in 2022).

Furthermore, comporting with the recent Brookings report [17], average estimates were higher than the conservative benchmarks noted in Table 4. Consider, for instance, the chances that a person who contracts COVID-19 recovers without medical intervention. Identifying the precise number is challenging because it depends on a patient's age, comorbidities, and reasons for hospital admission (COVID-19 symptoms or an incidental COVID-19 test). Nonetheless, those estimates typically range between 1% - 5% [17], not 30% (median) as suggested in this study, and there is no evidence that a healthy person under 65 years of age has around 12% chance (median) of ending up in ICU, as people's perceptions would suggest.

*Under*-estimation of negative consequences was less frequent but present. To be sure, the focus of this study was to document the over-estimation of COVID-19 risks and therefore contribute to the comparably more extensive literature on under-estimations and conspiracy beliefs [23, 26, 109, 131, 132]. However, as risk assessment literature suggests, both over- and under-estimations of risks may have consequences worth examining, and all erroneous estimates should be re-calibrated [4, 5, 18, 19, 133]. The most notable under-estimation type was the risk that COVID-19 poses to older citizens. Specifically, participants under-estimated the average age of death. In 2020 elderly care facilities indeed bore the brunt of all COVID-19 deaths, and in some countries, more than 70% of all people who died with COVID-19 were in aged care facilities [114, 115]. However, with greater awareness of COVID-19 dangers to the elderly, fewer deaths occurred in those facilities in subsequent waves [117]. Under-estimations were also evident when examining the alternative indicators. Consider, for instance, this erroneous claim: "if a vaccinated person tests positive for COVID-19, it means that the vaccine is not working". Evaluating this claim as true was related to lower NNP, but also lower self-report compliance with health-minded mandates. Importantly, beliefs in conspiracy theories (e.g., COVID-19 does not exist, PCR tests cannot tell between flu or COVID-19, COVID-19 has never been isolated) were also related to less NNP endorsement and less self-report compliance and non-vaccination status (Sample E).

## Risk estimation and NNP

Over-estimates of COVID-19 risks were generally related to stronger support of NNPs. Importantly, the relationship persisted even after considering the impact of C19 conspiracy theories, political ideology, and gender. However, it is crucial to note that while the core indicators predicted NNPs (evidenced in regression results where NNP was regressed on those indicators), not all of the seven indicators emerged as consistent predictors across samples (see S5–S8 Tables for details). Indicators that emerged as significant predictors include under-estimating recovery without medical intervention and over-estimating the risks of adverse outcomes for a healthy person under 65 (ICU, death, never recovering from COVID-19). Furthermore, while these core estimates predicted NNP (9-item policy variable), they did not consistently predict

the exploratory RN-fear (3-item assessment of fear of returning to 'normal'), suggesting that this variable might have different, unexamined antecedents.

## Perceived scientific consensus

Perceived scientific consensus pertaining to COVID-19 risks and its relationship with NNP support warrants special attention due to its unexpectedly strong and robust impact in Samples A-D, which I then further examined in Sample E (2022). People were more likely to support the NNPs not only if they believed there is scientific evidence on issues that are actually supported with scientific evidence (e.g., the efficacy of vaccines to prevent severe illnesses or deaths), but also on issues where the scientific consensus is not available, such as wearing masks while driving alone. Participants' responses were so consistent that all items–supported and unsupported–yielded a single, interpretable assessment. Despite this being a supplementary indicator (and thus was given to half of 1,200+ participants in 2021-based samples), it unexpectedly emerged as the single strongest predictor of NNPs over and above control variables and core indicators.

Sample E replicated these patterns with updated PSE items and showed that people who support NNPs believe that there is evidence behind statements such as N95 masks are safe for children's use during a typical school day, if the US had high (95%) vaccination rates, COVID-19 would have stopped spreading, and needing to prioritize children under the age of 12 for booster shots, even if there is no evidence behind those claims or WHO recommends otherwise (Table 15). These results suggest that it is possible that the PSE items are more related to *beliefs in scientific evidence*, which is a research question that my colleagues and I pursued in a different project [134].

## Alternative indicators (knowledge)

Because of the imperfections inherent in relying on quantitative- and percentage-based estimation as exclusive sources of COVID-19 risk perceptions, I sought to obtain a more nuanced understanding of these perceptions by introducing a series of indicators that assessed participants' knowledge of risks and risk mitigation tactics. I varied those across samples to minimize participant burden and increase the number of ways these questions can be asked. This overestimation of risk, and its corresponding relationship with continuing restrictions past vaccinations, also emerged when looking at an array of fact- or knowledge-based variables. For example, in Samples A–D, NNPs were endorsed more strongly by people who believed (incorrectly) that COVID-19 was the main cause of death in the US in 2020 or they labeled as false the UK practice that COVID-19 deaths are designated if the person dies within 28 days of a positive COVID-19 test.

## Other notable findings

I explored the relationship between risk estimations, personal characteristics, and COVID-19 mitigation behavior. One of the most consistent predictors of NNP was identifying as liberal. This relationship emerged not only in US samples but also in global Prolific ones and Australian/New Zealand community samples. Greater estimates of COVID-19 risks were also positively related to participants' current self-report compliance with health-minded measures, as evidenced by associations with core and PSE indicators. Conspiracy beliefs were negatively related to compliance. Finally, Sample E showed that having recovered from COVID-19 is negatively related to support for the *new normal*.

Collectively, these findings raise important questions about laypeople's perceptions of risks, COVID-19, and decision-makers' role in managing all misinformation. People's concern over

**Table 15. NNP and relationship with knowledge questions in Samples C and D.**

| Item | M (1–6) | SD | r with NNP | N | References |
|---|---|---|---|---|---|
| Sun rays can neutralize COVID-19. | **2.13** | 1.40 | -.426** | | **True** according to Ratnesar-Shumate, Williams [111] |
| In UK, if a person dies within 28 days of a positive COVID-19 test, they are counted as COVID-19 death (even if the person was in terminal stages of another illness). | **3.87** | 1.24 | -.386** | | **Factual statement–True** [112, 113]. |
| While accounting for a small portion of the population, people over 80 accounted for around half of all COVID-19 deaths. | **4.18** | 1.07 | -.170** | | **Generally true** [45]. When assessing COVID-19 risks, people generally under-estimate the risk to the elderly people. |
| In many Western countries, the majority of all COVID-19 deaths occurred in aged care facilities. | **4.14** | 1.06 | -.187** | | **Generally true** with the first wave of COVID-19 in Australia with 75% [114], Belgium with 61.3% [115], and Canada with more than 80% [116]. However, with greater awareness of aged care facilities and vaccines, this percentage is decreasing [117]. |
| People with obesity account for the majority of all COVID-19 hospitalizations. | **3.08** | 1.21 | -.161** | | **Generally true** [59, 135]. In addition, CDC notes: "Among 148,494 adults who received a COVID-19 diagnosis during an emergency department [ED] or inpatient visit at 238 U.S. hospitals during March–December 2020, 28.3% had overweight and 50.8% had obesity" [59]. |
| In 2020, COVID-19 was the main cause of death in the US. | **3.52** | 1.59 | .378** | | **Factual statement–False.** [118]. |
| If a vaccinated person tests positive for COVID-19, it means that the vaccine is not working. | **2.07** | 1.34 | -.362** | | **Factual statement–False.** CDC notes: "Based on what we know about vaccines for other diseases and early data from clinical trials, experts believe that getting a COVID-19 vaccine also helps keep you from getting seriously ill even if you do get COVID-19" [2]. |
| Mild asthma, sexually transmitted diseases [non-HIV]/AIDS, and severe acne put people in a high-risk category for COVID-19. | **2.81** | 1.40 | 0.04 | | **Factual statement–False.** [136]. However, moderate and severe asthma, and HIV/AIDS put people in a high-risk category for COVID-19 [137]. |
| The majority of COVID-19 spread is by people who are infected but show no symptoms (e.g., people with no fever or cough). | **4.03** | 1.32 | .258** | | **Mixed evidence** [70, 71, 74, 122–125]. |
| For healthy and fit people under 50 years of age, COVID-19 presents no greater risk than flu. | **2.98** | 1.58 | -.557** | | **Unclear** [117]. [37]. |
| Excess deaths are COVID-19 deaths (i.e., More people actually died of Covid than what is reported). | **3.71** | 1.45 | .452** | | **Unclear.** [138]: "Excess deaths not attributed to COVID-19 could reflect either immediate or delayed mortality from undocumented COVID-19 infection, or non–COVID-19 deaths secondary to the pandemic, such as from delayed care or behavioral health crises." |
| Long Covid symptoms include psychological and neurological disorders. | **4.01** | 1.39 | .143* | | **True** [56, 62, 66, 84, 128, 139]. |
| Most people recover completely within a few weeks. | **4.40** | 1.29 | -.279** | | **True.** Mayo Clinic notes: "Most people who have coronavirus disease 2019 (COVID-19) recover completely within a few weeks". Additional information: [54, 106] |
| A third of all people who test positive for COVID-19 will have lung scarring for at least 6 months. | **3.61** | 1.28 | .331** | | **False.** This statement represents a significant over-estimation of risk. There is no evidence that one-third of positive cases may have those symptoms for six months. |
| Long-term lung scarring and heart inflammation are unique to COVID-19 (i.e., not found in flu). | **3.54** | 1.34 | .240** | | **False.** Flu results in lung scarring and heart inflammation [37, 38]. Heart inflammation (inflammatory disease of the myocardium; myocarditis) often results from "common viral infections and post-viral immune-mediated responses"[130]. |

*Note.* The last four items pertain to 'Long Covid'.

COVID-19 can encourage pro-social behaviors to mitigate the risk of COVID-19 (e. g, contact tracing, compliance, and vaccination or vaccination intent). However, if their perceived risk and fear are disproportionally high (relative to the threat), it may become deleterious to cost-benefit analyses essential for COVID-19 and may be misused to encourage compliance based on fear. Laypeople have the power to impact public policy through the democratic process. If they greatly over-estimate COVID-19 risks and if they do not differentiate between practices that are based on evidence from those that are based on social contracts or desire to create a

feeling of safety [140], their feelings of short-term safety might greatly compromise long-term trust in experts, leaders, and science.

## Lessons, limitations, and boundaries of conclusions

When interpreting the results of this study, readers should be aware of several notable limitations. These limitations largely emerged due to efforts to employ multiple ways of asking participants about COVID-19 risks and interpreting those findings against the ever-changing knowledge of risks offered by health scientists.

First, while the study relied on multiple risk indicators, it is by no means a definitive record of all core risk information about COVID-19. Future research should expand these indicators and use the open data from this study to challenge and extend the field's knowledge of how laypeople perceive negative COVID-19 consequences. Similarly, future research should also consider multiple manifestations of the *new normal*, as I only focused on its darker side (vaccine passports and masking in perpetuity). However, through bringing attention to hygiene practices, the importance of strong healthcare systems and sick leave, and the management of other viral illnesses (including flu), there are numerous positive elements of post-pandemic life that might garner broader support from people.

Second, despite relying on participants from several different countries, the study did not examine the situational, cultural, or political factors that may explain the difference between, for instance, Australia/NZ (Sample D) and the US (Sample A). Sample differences are reported in S3 Table. Therefore, the converging results across those samples should only be used as evidence of generalizability; they should not be used to make inferences about cross-cultural comparisons or population-level attitudes. Follow-up research would benefit from documenting empirically the social conditions, government responses, and perceptions of media that lead to differences between country-level scores. Importantly, future research should also go beyond the cross-sectional data collection and examine whether risk estimation leads to *new normal* endorsement (or even vice versa) over time.

Third, the study only assessed risk estimation as one of many potential predictors of continuing restrictions post-vaccinations. While I chose this predictor deliberately based on recent evidence that people over-estimate COVID-19 risks, there are other reasons that could presumably influence participants' support for ongoing restrictions (e.g., participants' general pro-sociality, their perceptions of greater risk to the older adults, and trust in governments).

Fourth, findings from itemized knowledge-based indicators should be interpreted with caution. Recall that the main reason for supplementing core with alternative indicators was to expand the generalizability of the results while minimizing participant fatigue. Establishing the correctness and accuracy of COVID-19 items when knowledge continues to evolve is challenging, particularly for lay audiences. I selected items for which there is empirical support (e.g., vaccines reduce hospitalizations and deaths) and for which there is not (e.g., there is a high chance of COVID-19 transmission in outdoor settings). However, it is possible that there are other sources that can challenge the citations I provided in *Measures* section and that the available information changes. For instance, I had labeled the item claiming that the virus was made in the lab as a conspiracy theory (Samples A–D). Since that time, however, this possibility has been under investigation [110].

Those knowledge-based findings should, therefore, only be used to identify potential sources of over- or under-estimation, and this should be done while recognizing that the sources and the data represent what is known at one point in time. I decided to present those items and share this data in the main manuscript (vs. SI), despite the noted analytical limitations with the hope of understanding the sources of individuals' COVID-19 risk

miscalibration and allowing other researchers to do the same. Future research could simplify the interpretation by eliminating the continuous response structure, labeling each item as True/False, and employing an item response theory to assess participants' responses. The challenge here is that many COVID-19 claims come with caveats, but the benefits of binary assessments may outweigh the need for nuance. In retrospect, that is what I would have done with this study.

Fifth, I assessed participants' concerns over contracting COVID-19 in all samples. However, I inadequately explored its role because it is not evident whether its functions as a predictor or a dependent variable. To be sure, that challenge is present in all my other interpretations due to the cross-sectional nature of this data. From a theoretical perspective, however, it is plausible that concern over contracting COVID-19 leads one to consume fear-based information selectively, causing them to over-inflate the risks; or, it is equally plausible that over-inflation of risks causes one to be more concerned. Path analyses support both speculations (available from the author). Because the nature of the two variables is likely mutually reinforcing [141], future research can acknowledge that concern over COVID-19 has value and consider how it relates to risk-estimation and NNP endorsement over time.

Finally, the data and conclusions in this study should only be used with the intent of advancing one's understanding of laypeople's risk perceptions. While my study shows that over-estimations of COVID-19 risks are associated with NNP endorsement, I did not identify all the possible reasons why a person might support NNPs. Therefore, these findings should not be used to negatively judge individuals who choose to continue following the health-minded suggestions, wearing masks, and socially distancing. People who are cautious likely have valid reasons to continue following the public health recommendations, and those reasons were not documented in this study.

## Beyond the data: Unanswered questions and consideration of social implications of COVID-19 risk over-estimations

What should be done with the findings presented in this manuscript? At times when health institutes, news outlets, and governments seek to educate the population and warn them against conspiracy theories which often greatly *under*-estimate, if not fully negate COVID-19 [26, 131], results of this study and recent polls suggest that many people *over*-estimate risks. In turn, people who over-estimate risks are more likely to support the 'new normal' of continued or re-imposed restrictions. Deciding whether to scrutinize and correct misinformation associated with over-estimates requires considering the costs of doing and not doing so. If only under-estimates are corrected, people may continue to over-estimate the risks of COVID-19 and comply with mandates, perhaps in perpetuity. The collateral cost here is that people's actions may be driven by fear, making them susceptible to 'psychological shock tactics,' a tool admittedly used by the Belgian health minister [142]. If both erroneous estimates are corrected, people may start reducing their compliance with re-imposed restrictions on movement and refusing vaccines, boosters, and masks.

Determining whether general compliance (or resistance) is proportional to COVID-19's threat requires an open discussion of value systems, which in turn requires a decently informed and knowledgeable constituent base. Both over-, but also under-estimating risks can be costly in terms of time, money, resources, and even lives [4], and pandemics conditions are ripe for a medical version of the 'Hobbesian nightmare–the war against all' [143]. Similarly, Baral and colleagues [13] state that: "Minimizing deaths from COVID-19 over the long-term is critical, but so too is minimizing all-cause mortality and the preservation of other health and social services. Pandemics present no winners."

Based on the results of this study and the literature summarized in the introduction, uncorrected misinformation that inflates COVID-19 risks and presents data partially (e.g., featuring 'long Covid' articles without mentioning that the studies generating knowledge were conducted on hospitalized patients) will jeopardize risk assessment and will cast doubt that the existing fear-based availability cascades will stop with vaccinations and even with booster mandates. Instead, people and decision-makers may continue to apply the *maximin* principle for the foreseeable future and deal with all new and inevitable COVID-19 risks from the perspective of the worst-case scenario, thus preventing the restoration of social and economic life (WHO).

Therefore, I recommend that governments, decision-makers, and citizens confront all misinformation with equal rigor and hold media and public health figures accountable for educating rather than 'shocking' their constituents into compliance. After all, COVID-19 will likely continue [7], and public health education will be necessary to reduce risks to lives and health systems. Still, risk management should ensure that efforts to protect those who are vulnerable now do not come at the cost of those who will be vulnerable later (e.g., children whose education has been disrupted, young adults whose economic prospects may be jeopardized, and non-COVID-19 patients whose treatments have been disrupted). Despite their flaws and imperfections, democracies are stronger if decisions are transparent and citizens are reasonably well-informed [4], and the COVID-19 decision-making process should not be exempt from those principles [12]. Global policies based on selective dissemination and consumption of fear-based information and the pursuit of one objective at the expense of all others prevent the construction of a stable foundation on which lasting, empirically informed, and perhaps even more adaptable, post-pandemic life can be built.

## Supporting information

**S1 Fig. Visual distribution of core estimation indicators (Samples A–D).**
(PDF)

**S1 Table. NNP endorsement descriptive statistics by sample.**
(PDF)

**S2 Table. Additional descriptive information (Samples A–D).**
(PDF)

**S3 Table. ANOVA results.**
(PDF)

**S4 Table. One sample T test results per sample.**
(PDF)

**S5 Table. Correlations and descriptives per sample.**
(PDF)

**S6 Table. Regression results per sample.**
(PDF)

**S7 Table. Regression results with controls per sample.**
(PDF)

**S8 Table. Predicting RN-fear with controls.**
(PDF)

**S9 Table. Descriptive statistics and correlations without outliers.**
(PDF)

**S1 File. Additional results file.**
(PDF)

## Author Contributions

**Conceptualization:** Maja Graso.

**Data curation:** Maja Graso.

**Formal analysis:** Maja Graso.

**Funding acquisition:** Maja Graso.

**Investigation:** Maja Graso.

**Methodology:** Maja Graso.

**Project administration:** Maja Graso.

**Resources:** Maja Graso.

**Software:** Maja Graso.

**Writing – original draft:** Maja Graso.

**Writing – review & editing:** Maja Graso.

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
