## [Decision Letter · Decision Letter 0]

17 Jan 2022

PONE-D-21-17141

Covid-19 Risk Estimation Predicts Support for Continuing Restrictions Past Vaccinations

PLOS ONE

Dear Dr. Graso,

Thank you for submitting your manuscript to PLOS ONE. After careful consideration, we feel that it has merit but does not fully meet PLOS ONE’s publication criteria as it currently stands. Therefore, we invite you to submit a revised version of the manuscript that addresses the points raised during the review process.

Please see the reviewer reports below. The reviewers have asked for more detail in the methodology and motivations, as well as made suggestions for how to frame the study. Please also ensure that the manuscript is copyedited upon resubmission.

We look forward to receiving your revised manuscript.

Kind regards,

Hanna Landenmark

Senior Editor

PLOS ONE

https://journals.plos.org/plosone/s/file?id=ba62/PLOSOne_formatting_sample_title_authors_affiliations.pdf”

2. In your Methods section, please provide a justification for the sample size used in your study, including any relevant power calculations (if applicable)

Furthermore, please include additional information regarding the survey or questionnaire used in the study and ensure that you have provided sufficient details that others could replicate the analyses. For instance, if you developed a questionnaire as part of this study and it is not under a copyright more restrictive than CC-BY, please include a copy, in both the original language and English, as Supporting Information.

“Yes. This study is funded by the University of Otago's internal grant system (University of Otago Research Grant).”

“No competing interests.”

Reviewers' comments:

Reviewer's Responses to Questions

**Comments to the Author**

1. Is the manuscript technically sound, and do the data support the conclusions?

Reviewer #1: Yes

Reviewer #2: Yes

Reviewer #3: Partly

2. Has the statistical analysis been performed appropriately and rigorously? 

Reviewer #1: Yes

Reviewer #2: Yes

Reviewer #3: Yes

3. Have the authors made all data underlying the findings in their manuscript fully available?

Reviewer #1: Yes

Reviewer #2: Yes

Reviewer #3: Yes

4. Is the manuscript presented in an intelligible fashion and written in standard English?

Reviewer #1: Yes

Reviewer #2: Yes

Reviewer #3: Yes

5. Review Comments to the Author

Reviewer #1: The article clearly defines the problem it intends to analyze. However, the review of the investigations previously conducted returns a state-of-the-art too partial. In my humble opinion, the authors do not clearly identify the contradictions and inconsistencies present in the literature on the subject, ending up by suggesting in a biased and excessively monocausal way the possible steps to solve the problem.

There is now an enormous number of studies on the most varied repercussions that the current pandemic is having on the global population, as some of the examples below testify.

Okabe-Miyamoto, K., Folk, D., Lyubomirsky, S., & Dunn, E. W. (2021). Changes in social connection during COVID-19 social distancing: It’s not (household) size that matters, it’s who you’re with. Plos one, 16(1), e0245009. https://doi.org/10.1371/journal.pone.0245009

Luchetti, M. , Lee J.H. , Aschwanden, D. , Sesker , A., Strickhouser , J., Terracciano, A. & Sutin, A. (2020) . The trajectory of loneliness in response to COVID-19. American Psychologist. Advanceonlinepublication.http://dx.doi.org/10.1037/amp0000690

Tull, M. T., Edmonds, K. A., Scamaldo, K. M., Richmond, J. R., Rose, J. P., & Gratz, K. L. (2020). Psychological outcomes associated with stay-at-home orders and the perceived impact of COVID-19 on daily life. Psychiatry Research, 289, https://doi.org/10.1016/j.psychres.2020.113098

Even approaches different from attitudinal, psychometric and HIP ones, such as the special issue mentioned below, testify that the impact of the pandemic is multifaceted and impossible to reduce to a mere problem of “accurate information”. While recognizing the central role that the society cannot be simplistically reduced to a source of information, but must be considered a source of meaning. On issues of their interest, people construct questions and search for answers, rather than merely perceiving and processing information derived from the social context. See for example:

Papers on Social Representations - Special issue (2020). Social Representations of Covid-19: Rethinking the Pandemic’s Reality and Social Representations [Vol 29, No 2]

The article certainly proposes an interesting vision of the impact of the current pandemic on the population, imagining an extremely original way to help people to cope with it, however, my impression is that the author has a vision of some dimensions of the problem too unilinear.

In general, it seems to me that the paper technically sounds, that the claims are convincing, appropriately discussed in the context of previous literature and fully supported by the empirical data. Overall, the paper is very well organized, clearly written and sharp in its observations regarding the consequences of the subject of study.

Reviewer #2: This manuscript – essentially exploring extent to which people’s estimation of COVID-19 risks influences their support for living with restrictions in the new normal – is one of the most enjoyable pieces of scientific writing I have reviewed since the start of the pandemic. As such, the study is built on a thoughtful rationale and a sound base in theory. The methodological approach is rigorous (but not without its limitations that have been sufficiently addressed in the discussion) to the extent to which the research questions are answered, and is explained in sufficient detail to merit replication. The discussion focuses on the key findings and while some of the points are mildly provocative (which makes this work even more engaging), the manuscript is careful in its claims without extrapolating beyond the scope of the data. Overall, I am pleased to recommend this manuscript for publication should the very minor amendments listed below be made.

1) There’s a missing word in this sentence (pg. 10, last paragraph): “The secondary measure of (?) was a 3-item affect-based variable reflecting…”

2) In Table 4, the asterisk is not qualified

3) Its not clear why the Cronbach’s alpha scores for the RN-Fear scale have not been presented.

4) Table 5: PSC needs to be elaborated.

5) While the authors were mindful to ask participants about “Concern over Contracting COVID-19”, it is unclear why they were not asked directly about whether they had previously contracted COVID-19 or not? Disease experience/disease history could be a critical factor in shaping risk perceptions around the ‘new normal’.

6) In pg. 16, the word ‘limitations’ is spelt without the ‘o’.

7) In pg. 19, it needs to be changed to “both 5-item scales” (plural)

8) In pg. 34, the sentence reads “only if they believed there is scientific evidence on issues that are actually supported with scientific evidence”. Should the second instance of evidence be changed to “consensus” in line with the arguments in this portion?

9) In pg. 34, the first and second paragraphs need to be written a bit more clearly. In the first para, the sentence starting with “People were more likely…”, is the suggestion that people believe in scientific evidence even in some instances where scientific consensus is not available? In the second para, am I right that in line 5 (sentence starting with “However...”), risk needs to be qualified as “perceived risk”? Also, perhaps change to “lay people have the power” from “to power”?

10) The last, and only major point to made pertains to the discussion around the role public perceptions of scientific consensus. There are three comments in relation to this. First, the aspect of scientific consensus arrives a bit unexpectedly in the methods without being incorporated sufficiently into the background. This would be helpful to do by way of a setting a stage for this variable especially as it seems to form an integral part of your discussion section. Second, while the author is rightfully concerned about ‘false consensus’, it would be worthwhile to bear in mind that this kind of ‘false consensus’ might have been arrived at over time after repeated exposure to excessive amounts of media coverage, interpersonal communication with trusted social networks, personal experience with COVID-19 and other engagements with the informational environment. As such, it would be useful to qualify the concern expressed by the author by contextualising it against the informational environment (or, more specifically, the infodemic), lest it might seem like the ‘false consensus’ is formed in a vacuum. Third, While it is appreciated that the discussion does not overstep its remit, it would nonetheless be useful to include a brief discussion practical implications of these findings for risk communication practice and policy.

Reviewer #3: • I would consider changing the title to make it immediately clear that the paper focuses on individuals’ own estimation of COVID-19 risks. The phrase “COVID-19 risk estimation” is a little ambiguous; perhaps something like “individuals’ COVID-19 risk perceptions” would be clearer.

• The paper has several typos and grammatical mistakes. More editing is needed.

• Did you collect information about which countries participants in the Prolific international sample lived in? This information would be useful in considering how responses from those samples compare to responses from the other samples.

• Were the other attention check measures similar to the sample item included in the paper (“For quality control, please select ‘3’”). If so, the substantial proportion of people who failed the attention check measures is concerning, particularly in Sample C (15% of participants in Sample C failed). Can the author comment on this?

• One of the items capturing support for new normal practices in Table 3 is “Implement a program similar to COVID-19.” Is this item missing a word or does it contain a typo? I don’t understand what it means.

• What does the shading of certain rows in Table 5 and Table 6 indicate? This should be explained in the table notes.

• Can the author comment on why they chose not to make a scale using the core risk-estimation indicators and instead included them as separate predictors in the regression model? Given that the items are correlated with one another, including them all in the model may have resulted in variance inflation. Additionally, given the number of correlated variables included as predictors in the model, you should test the joint significance of the core risk-estimation indicators rather than looking at the individual correlation coefficients.

• There is a sentence fragment on page 24 of the manuscript: “However, what makes the label ‘over-estimation’ a more appropriate descriptor as opposed to ‘higher estimation’.”

• In general, I think the author may just be trying to fit too much into one paper. With so many different variables and analyses to describe, I found the paper difficult to follow. I think tightening the organization of the paper and being as concise as possible could help with this. It would also be helpful to lay out the paper’s main hypotheses more explicitly are and describe the analyses used to test each one.

• In my view, the major limitation of this paper is its cross-sectional nature. Using this data, it is not possible to say whether over-estimation of COVID risks leads to ‘new normal’ policy support or vice versa. The author should discuss the possibility of reverse causation and its implications in the paper.

6. PLOS authors have the option to publish the peer review history of their article (what does this mean?). If published, this will include your full peer review and any attached files.

Reviewer #1: **Yes: **Roberto Fasanelli

Reviewer #2: No

Reviewer #3: No

---

## [Author Response · Author response to Decision Letter 0]

3 Feb 2022

Please see my detailed responses to editors and reviewers attached as a separate document. Thank you.

---

## [Editor Report · Decision Letter 1]

28 Feb 2022

PONE-D-21-17141R1Covid-19 Risk Perceptions, Over-estimations Predict Support for the ‘New Normal’ and Continuing Restrictions Past VaccinationsPLOS ONE

Dear Dr. Graso,

Thank you for submitting your manuscript to PLOS ONE. After careful consideration, we feel that it has merit but does not fully meet PLOS ONE’s publication criteria as it currently stands. Therefore, we invite you to submit a revised version of the manuscript that addresses the points raised during the review process.

We look forward to receiving your revised manuscript.

Kind regards,

Santosh Vijaykumar

Academic Editor

PLOS ONE

Journal Requirements:

Additional Editor Comments :

Dear Dr. Maja Graso,

Thank you for submitting your manuscript to PLOS ONE. After careful consideration, I believe that you have satisfactorily addressed the theoretical, methodological and typographical concerns raised by the reviewers on the first version of your submission. As such, the manuscript stands substantially improved in terms of its focus and clarity with the way in which the findings have now been divvied and organised between the main manuscript and supplementary online materials. Given these improvements, we are pleased to invite you to resubmit the manuscript after addressing the following minor comments. In the interest of scientific transparency, kindly note that I participated as a reviewer for the initial evaluation of this manuscript.

1. Throughout the manuscript, please replace Covid-19 with COVID-19 (all caps)

2. Introduction, pg. 4, amend (i.e., deaths of Covid-19 or hospitalizations) to (i.e., reduction in deaths of Covid-19 or hospitalizations)

3. Introduction, pg. 4, 3rd para, 3rd line: replace "underestimation of COVID-19" with "under-estimation of the threat of COVID-19" (or any other qualification you feel might be more appropriate)

4. Pg. 5, end of first para, please qualify the term "informational and reputational cascades" and the "moralization of COVID-19".

5. In Table 4, for #1 it is not clear what the estimate should be, as it has been provided for the other statements/questions.

6. In Table 5, the word 'consensus' is misspelt in items #1 and #3.

7. In pg. 21, 2nd para, line 6, there's a phrase: "aggregating items that less empirical support" which needs correction.

8. In pg. 28, it'd be useful to provide a brief description of the "lab leak hypothesis" for the benefit of the readers who might not be familiar with this issue.

9. In pg. 31, the phrase: "SI provides additional data based 2021 samples" needs to changed to "...based ON 2021 samples"

Best wishes,

Santosh
---

## [Author Response · Author response to Decision Letter 1]

1 Mar 2022

Please see the attached response file.

In summary, I made the following changes: 

1. Throughout the manuscript, please replace Covid-19 with COVID-19 (all caps)

a. Done. I made all requested changes to COVID-19 term.

2. Introduction, pg. 4, amend (i.e., deaths of Covid-19 or hospitalizations) to (i.e., reduction in deaths of Covid-19 or hospitalizations).

a. Done. I made the following adjustment:

“Whether to continue restrictions is not a decision that can be made exclusively by health scientists because societal well-being cannot be reduced to a single indicator of success (i.e., reduction in deaths of COVID-19 or hospitalizations); instead, it involves consideration of competing priorities and limited resources (mental health, economics, and education, among others) (11, 12).

3. Introduction, pg. 4, 3rd para, 3rd line: replace "underestimation of COVID-19" with "under-estimation of the threat of COVID-19" (or any other qualification you feel might be more appropriate).

a. Done. I made the following adjustment:

“For instance, under-estimation of the threat of COVID-19 or belief in conspiracy theories (e.g., unfounded cures) may lead people to disregard health-minded rules and risk the lives of the vulnerable (13-15).”

4. Pg. 5, end of first para, please qualify the term "informational and reputational cascades" and the "moralization of COVID-19".

Thank you for encouraging me to be more precise. I did not qualify those statements in this statement, only because I do so in the subsequent section “Origins and Catalysts of Risk Over-estimation: Theoretical Foundation”. I have now refined the definitions and state the following: 

 “I proceed to explain how my predictions are informed by complementary theories on perceptions of unknown (vs. known) risks (18), availability and reputational cascades (4, 5, 17, 18), and moralization of COVID-19 (19, 20).”

 The following section defines these terms:

 “While erring on the side of extreme caution is defensible in the absence of information, continuing to do so in light of new information is not. This fear-based focus can further perpetuate availability cascades where new information is not used to revisit the cost-benefit analysis but is selectively disseminated and censored; information that deviates from the narrative may elicit reputational damages or moral outrage (20, 21, 28). These forces may also be explained with moralization, a process by which an attitude becomes a matter of moral imperative (20). When an attitude becomes moralized, it becomes absolute, intolerant, and resistant to change, which may further perpetuate the availability cascade of moralized information.” 

5. In Table 4, for #1 it is not clear what the estimate should be, as it has been provided for the other statements/questions.

a. Thank you. May I clarify that you are referring to the Core Indicator 1 (What is the average age of a person who died of COVID-19 in the US?)? If so, the estimation benchmark is 78+ (noted in Table 4):

“The average age for deceased COVID-19 patients tends to be approximately 80 years of age, with some variation based on gender and reporting protocols (36, 45-51).”

6. In Table 5, the word 'consensus' is misspelt in items #1 and #3.

a. Done. Thank you for bringing this error to my attention. I edited my manuscript one more time and detected a few similar errors in ‘consensus’.

7. In pg. 21, 2nd para, line 6, there's a phrase: "aggregating items that less empirical support" which needs correction.

a. Thank you for bringing this error to my attention. The statement lacked ‘that’. Accordingly, I changed it to: “I considered evaluating the data using a 5-item scale aggregating items that have less empirical support (PSE –supported; α = .78), and a 5-item scale aggregating items that have more empirical support (PSE – unsupported; α = .86).”

8. In pg. 28, it'd be useful to provide a brief description of the "lab leak hypothesis" for the benefit of the readers who might not be familiar with this issue.

a. Done. I now changed the description to the following: Since the data collection (April, 2021) and while the manuscript was under review, the so-called “lab leak hypothesis” (i.e., belief that COVID-19 emerged from a lab) has received more attention and according to Maxmen and Mallapaty (106), many research institutes are calling for a deeper investigation of this possibility.

9. In pg. 31, the phrase: "SI provides additional data based 2021 samples" needs to changed to "...based ON 2021 samples"

a. Done. The line now states “SI provides additional data based on 2021 samples, including a visual representation of frequency distributions, which can aid readers in evaluating this work.”

Besides these changes, I also made the following minor adjustments:

1. I reported median estimation scores, in addition to mean and standard deviation.

2. I replaced the image-based tables with Word-based tables. The actual numbers remained the same, but the Word file is cleaner.

3. I edited the manuscript for clarity and precision.

Thank you once again for your continuing support. I look forward to your feedback.

~ Maja Graso

University of Otago

New Zealand

---

## [Editor Report · Decision Letter 2]

24 Mar 2022

The New Normal: Covid-19 Risk Perceptions and Support for Continuing Restrictions Past Vaccinations”

PONE-D-21-17141R2

Dear Dr. Graso,

We’re pleased to inform you that your manuscript has been judged scientifically suitable for publication and will be formally accepted for publication once it meets all outstanding technical requirements.

Kind regards,

Santosh Vijaykumar

Guest Editor

PLOS ONE

Additional Editor Comments (optional):

Dear Dr. Maja Graso,

I am pleased to recommend acceptance of this article in its current form given the satisfactory responses to all comments and suggestions. I wish you all the very best moving forward and trust that the Editor will be in touch with you as regards next steps.

Best wishes,

Santosh
---

## [Editor Report · Acceptance letter]

31 Mar 2022

PONE-D-21-17141R2 

The New Normal: Covid-19 Risk Perceptions and Support for Continuing Restrictions Past Vaccinations 

Dear Dr. Graso:

I'm pleased to inform you that your manuscript has been deemed suitable for publication in PLOS ONE. Congratulations! Your manuscript is now with our production department. 

Kind regards, 

on behalf of

Dr. Santosh Vijaykumar 

Guest Editor

PLOS ONE